# A Clean Slate for Offline RL

**Matthew T. Jackson**[*]   **Uljad Berdica**[*]   **Jarek Liesen**[*]
**Shimon Whiteson**   **Jakob N. Foerster**

University of Oxford
{jackson,uljadb,jarek}@robots.ox.ac.uk

## Abstract

Progress in offline reinforcement learning (RL) has been impeded by ambiguous problem definitions and entangled algorithmic designs, resulting in inconsistent implementations, insufficient ablations, and unfair evaluations. Although offline RL explicitly avoids environment interaction, prior methods frequently employ extensive, undocumented online evaluation for hyperparameter tuning, complicating method comparisons. Moreover, existing reference implementations differ significantly in boilerplate code, obscuring their core algorithmic contributions. We address these challenges by first introducing a rigorous taxonomy and a transparent evaluation procedure that explicitly quantifies online tuning budgets. To resolve opaque algorithmic design, we provide clean, minimalistic, single-file implementations of various model-free and model-based offline RL methods, significantly enhancing clarity and achieving substantial speed-ups. Leveraging these streamlined implementations, we propose *Unifloral*, a unified algorithm that encapsulates diverse prior approaches within a single, comprehensive hyperparameter space, enabling algorithm development in a shared hyperparameter space. Using Unifloral with our rigorous evaluation procedure, we develop two novel algorithms—TD3-AWR (model-free) and MoBRAC (model-based)—which substantially outperform established baselines. All code for this project can be found in our public codebase.

## 1  Introduction

Offline reinforcement learning (RL)—the task of learning effective policies from pre-collected, static datasets—is critical for applying RL in real-world settings where online experimentation is expensive or risky. Despite significant interest [1–5], the field has struggled to converge on clear, actionable insights. Algorithms and methods proliferate rapidly but no broadly agreed-upon conclusions or standardized benchmarks have emerged [6]. This undermines both practical application and theoretical progress. In this work, we identify and address two primary problems that contribute to stagnation and confusion in offline RL research: an ambiguous problem setting and opaque algorithmic design.

**Problem 1: Ambiguous Problem Setting**   Recent work in offline RL has lacked a rigorously articulated definition or standardized evaluation procedure. The broad mission statement, learning from a static dataset without direct environment interaction, is prone to misinterpretation that skews proposed methods towards impractical evaluation practices. Existing literature implicitly relaxes various definitions concerning critical details such as hyperparameter tuning allowances [4, 7], the extent of post-deployment policy adaptation [8], and the specifics of evaluation procedures [9]. Consequently, comparisons between methods are confounded as each study might assume fundamentally different

---

[*]Equal contribution.

39th Conference on Neural Information Processing Systems (NeurIPS 2025).

experimental conditions. While some approaches restrict tuning based on related dataset performance [10], most approaches extensively tune hyperparameters on the target environment [11, 5, 12]. Using the target environment to tune hyperparameters needs a large number of *online* evaluations, which is in conflict with the basic premise of *offline* RL.

**Solution 1: A Novel Taxonomy and Evaluation Procedure**   We first introduce a rigorous and explicit taxonomy of offline RL evaluation variants (Section 3.1) and specify the one we find to be implicitly adopted by most prior research. To facilitate consistent and transparent evaluation, we propose a rigorous procedure for this setting (Section 3.2) that evaluates algorithmic performance using a fixed hyperparameter range across multiple datasets. This procedure explicitly quantifies performance at various permissible levels of online hyperparameter tuning, i.e., interactions with the *target* environment, thus providing clarity about the practical deployment requirements of each method. To ensure ease of adoption and reproducibility, we release a straightforward software interface for performing this evaluation procedure, thereby empowering future work to evaluate offline RL algorithms robustly and transparently.

**Problem 2: Opaque Algorithmic Design**   Offline RL methods are often presented as intricate bundles with intertwining algorithmic components, implementation-specific details, and unclear tuning procedures. Researchers compare proposed methods to baseline performance quoted directly from prior publications [6], inadvertently propagating these methodological issues. As a result, it is difficult to isolate the impact of individual methodological choices. Thus, the state-of-the-art remains ambiguous, with no method demonstrating uniformly strong performance across all datasets [13–15].

**Solution 2: Consistent Reimplementations and a Unified Algorithm**   We first dissect the novel components of prior algorithms by defining a phylogenetic tree based on their compositional structure (Section 4.1). We use this representation to provide single-file *reimplementations* of a wide range of offline RL methods. These minimal implementations eliminate extraneous code differences and highlight fundamental components, as well as achieving average training speedups of $131.5\times$ and $74.8\times$ against OfflineRL-Kit [16] and CORL [17], two leading offline RL libraries. Furthermore, we propose a unified offline RL algorithm (**Unifloral**, Section 4.2) that integrates core components from various prior methods into one coherent framework. Crucially, Unifloral provides a single, *unified* hyperparameter space containing all of these algorithms.

Leveraging Unifloral with our evaluation procedure, we introduce two novel offline RL methods: a model-free approach (**TD3-AWR**, Section 5.1) and a model-based one (**MoBRAC**, Section 5.2). These methods demonstrate substantial performance improvements over established baselines, validating both our unified methodology and rigorous evaluation framework.

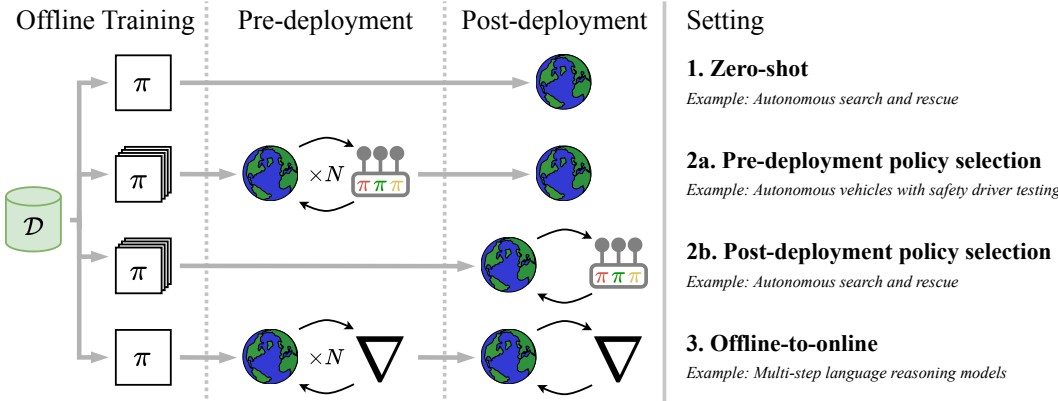

Figure 1: Formalizing the variants of offline RL—we define a range of offline RL variants (Section 3.1), with policy performance being measured post-deployment. Pre-deployment policy selection (2a) and post-deployment policy selection (2b) use a policy-selection bandit after offline training, whilst (3) uses unrestricted policy updates.

## 2 Preliminaries

### 2.1 Reinforcement Learning

We apply RL to a finite-horizon Markov Decision Process (MDP) defined by the tuple $\langle S_0, \mathcal{S}, \mathcal{A}, D, R, T \rangle$. Here $\mathcal{S}$ is the state space, $\mathcal{A}$ is the action space, and $T$ is the horizon. $D : \mathcal{S} \times \mathcal{A} \to \Delta(\mathcal{S})$ is the transition dynamics, defining how the state changes given a state and the action taken on that state. $\Delta(\mathcal{S})$ is the set of all possible distributions over $s$. The scalar reward function is $R : \mathcal{S} \times \mathcal{A} \to \mathbb{R}$. The environments in this paper are all fully observable as the Markov state is directly observed at each timestep.

A policy $\pi$ maps a state in $\mathcal{S}$ to an action distribution over $\mathcal{A}$. The policy is trained to maximize the expected return $J_M^\pi$ for a given MDP $M$ with trajectory length $T$:

$$J_M^\pi := \mathbb{E}_{a_{0:T} \sim \pi, s_0 \sim \mathcal{S}_0, s_{1:T} \sim D} \left[ \sum_{t=0}^{T} r_t \right]. \tag{1}$$

### 2.2 Offline Reinforcement Learning

Offline RL methods use a pre-collected dataset to optimize a *target* policy to maximize $J^\pi$ by, without online interactions in the environment. This dataset consists of transitions $(s_i, a_i, r_i, s_{i+1}, a_{i+1})$ for $i = 1, \ldots, N$, where $s_i, s_{i+1} \in \mathcal{S}, a_i \in \mathcal{A}, r_i \in \mathbb{R}$ are the current and next states, action, and reward, respectively. Here, initial states are drawn from the distribution $s_0 \sim \mathcal{S}_0$ and trajectories are gathered through a *behaviour* policy $\pi_b$ interacting with the environment. Since $\pi_b$ may be suboptimal, the resulting dataset might not contain sufficient coverage of the environment's state space to learn an effective policy.

An effective offline RL method must learn policies that generalize from this limited dataset to perform reliably when deployed in their environment. Typically, these methods require significant regularization to avoid overestimation bias. For model-free methods, this is commonly done with critic ensembles, where the minimum state value estimated by the ensemble is used for policy optimization. Model-based methods generalize by training a dynamics model $\hat{D}(s, a)$ to predict future states and rewards. This can be used to generate synthetic rollouts from the target policy, allowing for direct optimization of its performance.

## 3 Refining Evaluation in Offline RL

This section describes our taxonomy of offline RL, illustrated in Figure 1, which motivates our evaluation procedure in Figure 2. We also outline the procedure in detail and use it to analyze the performance of a set of model-free and model-based algorithms in multiple environments.

### 3.1 Variants of Offline RL

The goal of offline RL is to train an agent using solely offline data, with the objective of maximizing performance from *deployment*, i.e., the point where the agent is evaluated online. In this setting, deployment marks a strict separation between the offline training phase and the online evaluation phase. However, some methods may relax this strict separation in two ways. Firstly, *pre-deployment interaction* allows the agent to take limited interactions with the environment before deployment to improve post-deployment performance. For instance, to tune hyperparameters before selecting a policy for deployment. Secondly, *post-deployment adaptation* allows the agent to continue learning after deployment, and the performance metric includes all returns collected after deployment. Examples include dataset aggregation from multiple online episodes [18], selection from a set of policies trained offline [7, 9], and fine-tuning a single policy [8], all of which can be performed both before and after deployment. While any combination of these is possible, we identify four key settings.

> **A Taxonomy of Offline RL**
>
> 1. ZERO-SHOT OFFLINE RL
>    - Train **one policy** offline, then deploy online with no further adaptation.
>    - No pre-deployment interaction, no post-deployment adaptation.
> 2a. OFFLINE RL WITH PRE-DEPLOYMENT ONLINE POLICY SELECTION
>    - Train a **set of policies** offline, select the best policy based on $N$ online evaluations before deployment.
>    - Limited pre-deployment interaction, no post-deployment adaptation.
> 2b. OFFLINE RL WITH POST-DEPLOYMENT ONLINE POLICY SELECTION
>    - Train a **set of policies** offline, then deploy online, adaptively selecting a policy every episode based on online performance.
>    - No pre-deployment interaction, post-deployment adaptation via policy selection.
> 3. OFFLINE-TO-ONLINE RL
>    - Train **one policy** offline, then deploy online and fine-tune the policy on online data.
>    - Limited pre-deployment interaction and post-deployment adaptation via finetuning.

Many offline RL papers implicitly perform pre-deployment policy selection (Setting 2a), as they report final performance after extensive hyperparameter tuning involving online evaluation [11, 5]. However, due to differences in the number of hyperparameters or computational resources, this tuning process varies in scope across studies. As a result, *reported performances are often not directly comparable* since they reflect not only algorithmic quality but also differences in tuning budgets. Furthermore, these procedures typically assume low-variance estimates of each policy's performance, determined by an *indefinite number of online evaluations*. This is rarely made explicit as hyperparameter tuning is often considered a technical detail and not part of the method, even though it can dramatically affect performance (Section 3.3).

Finally, much prior work has blurred the line between algorithms and hyperparameters in offline RL, proposing different hyperparameter values or ranges for each task. This ambiguity enables the same "method" to have dramatically different behaviour across tasks,

> **A Definition of Offline RL Methods**
>
> A method in offline RL consists of an **algorithm** and a **fixed sampling range** for each hyperparameter.

undermining the assumption of limited interactions by essentially proposing a different method for each task. To resolve this, we define an offline RL method to include a fixed hyperparameter range, which remains constant across datasets (see A Definition of Offline RL Methods).

## 3.2 Proposed Evaluation Procedure

We now propose a rigorous and practical evaluation procedure for offline RL with pre-deployment policy selection (Setting 2a), as it implicitly is the standard setting for evaluating offline RL methods (see Section 3.1). Our goal is to evaluate offline RL algorithms under a fixed budget of $N$ pre-deployment environment interactions used for tuning. We measure this budget in terms of the number of evaluation episodes, reflecting practical deployment constraints where each online interaction can be costly. Whilst the tuning algorithm may be defined as part of the method, most research focuses on offline policy optimization prior to tuning. Therefore, we provide an upper confidence bound (UCB) bandit [19] in our implementation as the default tuning algorithm.

Furthermore, to reflect real-world limitations, we assume that the expected return of each policy is not directly observable, with each pull from the bandit sampling a *single* episodic return from that policy's return distribution. This models the high-variance, sample-limited setting typical in real deployments, where evaluating a policy's performance requires interacting with the environment and yields only noisy, episodic feedback. The importance of this is demonstrated by the emergence of distractor policies, as discussed in Section 3.3.

In essence, our evaluation procedure repeatedly simulates hyperparameter tuning with a fixed online budget, using a bandit to select a single policy for final deployment. This procedure (Figure 2) has two steps: score collection and bandit evaluation.

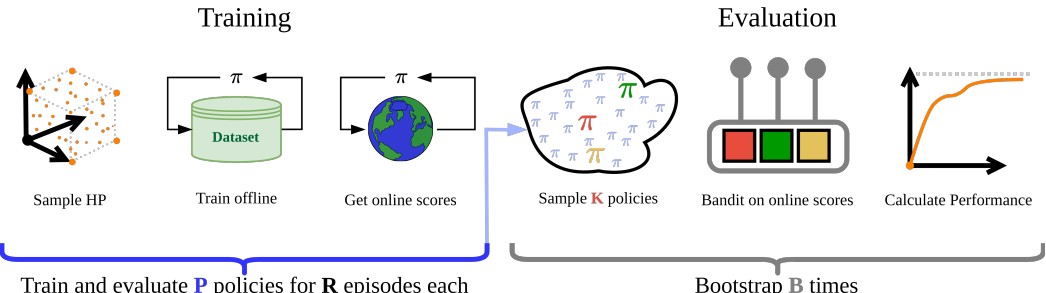

Figure 2: Overview of our evaluation procedure. Left: We sample hyperparameters, train the corresponding policies, and collect their final evaluation scores. Right: We simulate hyperparameter tuning using the collected scores by subsampling $K$ policy scores and recording the best-arm performance of a UCB tuning bandit operating over them.

**Step 1: Train Policies and Collect Scores**    Firstly, we collect a dataset of episodic evaluation scores from policies trained by the target algorithm. To do this, we sample $P$ hyperparameter settings (with replacement and random seeds) from the range defined by the method, and then train $P$ corresponding policies. These policies are evaluated online for a large number of episodes $R$ and their episodic scores recorded. Following this, the policies may be discarded as only their episodic scores are required for bandit evaluation.

**Step 2: Run Bootstrapped Tuning Bandit**    Using our collected episodic evaluation dataset, we then repeatedly simulate hyperparameter tuning to measure algorithm performance at different tuning budgets. This is performed by subsampling $K$ policies[1] (i.e., their corresponding episodic scores) and running a multi-armed bandit over them. In this bandit, each arm corresponds to a policy, with each pull sampling one episode's return from the corresponding policy. At each number of pulls $N$, we evaluate the performance of the algorithm by selecting the policy estimated to have the highest return by the bandit, and taking its true average return. We repeat this process $B$ times to obtain a bootstrapped estimate of algorithm performance.

**Recommended Datasets**    It is essential to evaluate methods on a diverse distribution of tasks to ensure generality. Alarmingly, the majority of offline RL methods considered in this work were evaluated *only* on MuJoCo and Adroit tasks from the D4RL suite [20]. While computational budgets may be limited, we argue that they would be better spent considering a wider range of tasks and behaviour policies. In order to make environment selection consistent, we recommend starting with the following environments, where algorithms currently obtain non-trivial performance: **hopper-medium**, **halfcheetah-medium-expert**, and **walker2d-medium-replay**, as a representative subset of MuJoCo locomotion; **pen-human**, **pen-cloned**, and **pen-expert**, as algorithms often achieve zero or perfect performance on other Adroit environments; **kitchen-mixed**, **maze2d-large**, and **antmaze-large-diverse**, to provide diversity in the evaluated environments.

### 3.3  Results

In Figure 3, we evaluate a range of prior algorithms (list in Appendix A). For this, we uniformly sample from the hyperparameter tuning ranges specified in each algorithm's original paper or the union of ranges when multiple are provided. Generally, an algorithm performs better if its curve is closer to the top left corner of a plot, representing strong performance after few online interactions. Prior work has typically reported performance after unlimited online tuning, which is the limit of the score with an increasing number of policy evaluations, i.e., the top right corner.

**Inconsistent Algorithm Performance**    No algorithm consistently performs well across all datasets. However, ReBRAC and IQL are competitive for the overall best performing algorithm, with ReBRAC achieving top performance at some number of evaluations on 5 out of 9 datasets and IQL on 4 out of 9 datasets. Even though both of these algorithms are worse than competing baselines on other datasets, we believe them to be the clearest baselines for future method development, as done in Section 5.1.

---

[1]We fix $K = 8$ in our experiments but encourage future evaluation under other values.

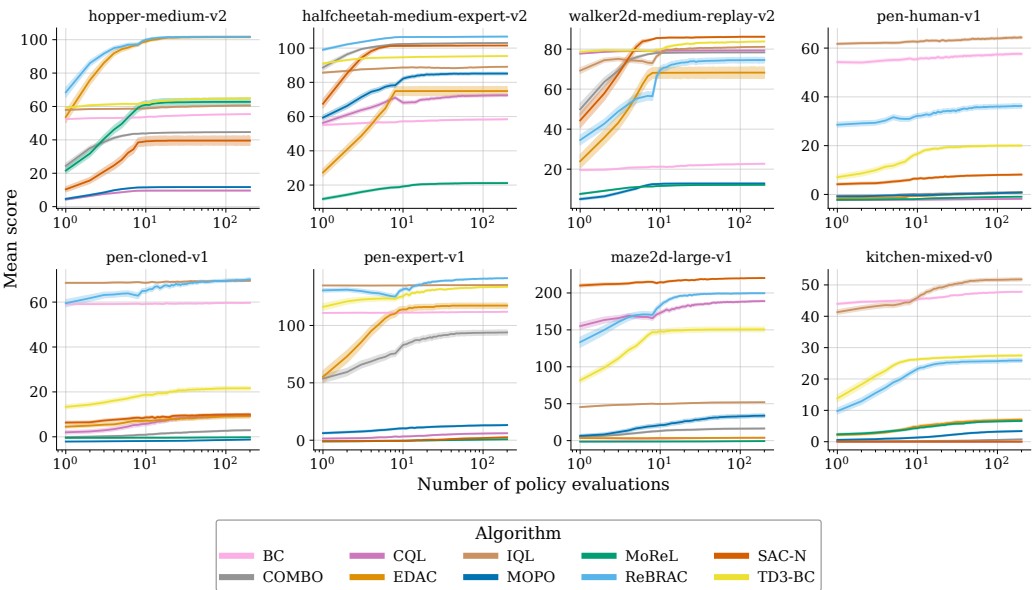

Figure 3: Evaluation of prior algorithms—mean and 95% CI over 500 bandit rollouts, with $K = 8$ policy arms subsampled from 20 trained policies each rollout. The $x$-axis denotes the number of bandit pulls, whilst the $y$-axis denotes the true expected score of the *estimated best arm* after $x$ pulls. The full evaluation results in Appendix B.

**Overfit Model-Based Methods** The model-based algorithms we evaluate—MOPO, MOReL, and COMBO (Appendix A.2)—achieve notably poor performance on all non-locomotion datasets, ranking no higher than 6[th] out of the 10 evaluated algorithms (and failing to beat BC) at any number of policy evaluations. While these results are surprising, we emphasize that our implementation successfully reproduces reference results with the specialized hyperparameters for each dataset (Appendix G). Instead, these results suggest that these methods are deeply overfit to the locomotion datasets they were originally evaluated on (Appendix C), providing a sobering reflection of the field.

**Distractor Policy Phenomenon** While performance typically improves as more bandit arms are pulled, certain performance curves exhibit distinctive dips—temporary decreases in measured performance despite additional policy evaluations. To better understand this, we examine the ranked performance distribution of numerous ReBRAC policies trained on hopper-medium (Figure 4). This analysis reveals a notable cluster of policies that exhibit suboptimal average performance but possess a *higher maximum performance* compared to consistently better-performing policies. We refer to these anomalous policies as *distractor policies*.

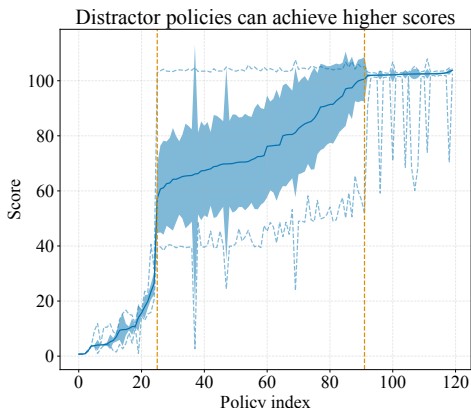

Figure 4: Ranked ReBRAC performance— blue shaded area, solid and dashed lines representing the standard deviation, mean, min, and max episodic return, respectively.

To demonstrate their impact on evaluation, we simulate the initial phase of a bandit rollout over these policies, i.e., when the bandit enumerates all arms (Figure 9a). Over this phase, we observe a clear *increase* in the probability of preferring a distractor policy, explaining the initial decrease in evaluation performance. This phenomenon runs counter to the expectation that increasing policy evaluations would monotonically reduce estimator variance and underscores the need to directly consider environment interactions in evaluation, a crucial distinction from prior evaluation methodologies [9]. Further analysis of distractor policies is provided in Appendix D.

## 4 Elucidating Algorithm Design in Offline RL

In this section, we seek to simplify algorithm design in offline RL. Firstly, we present a genealogy of prior algorithms, using it to propose and implement a set of *compositional* reimplementations. Following this, we propose a unified algorithm, Unifloral, capable of expressing these methods—as well as any combination of their components—in a single hyperparameter space.

### 4.1 Disentangling Prior Methods

New offline RL methods are typically derived from preceding ones by adding or editing individual components of the agent's objective or architecture. Despite this, methods typically suffer from a range of unnecessary implementation differences, making it difficult for researchers to identify their contribution or fairly compare methods. Even in popular single-file implementations, we observe significant code differences between "parent" and "child" algorithms, which should require only the individual components to be edited. This encourages researchers to compare entire algorithms rather than ablating components. We discuss this and how it informs our code philosophy in Appendix E.

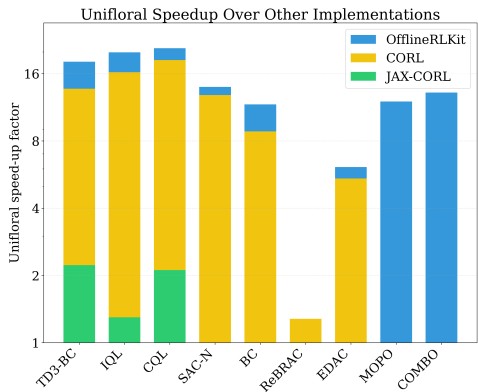

Figure 5: Speed up from our JAX reimplementations — algorithms trained for 1M update steps on `HalfCheetah-medium-expert` using a single L40S GPU. Our library, Unifloral, is the fastest across the board. Full details can be found in Appendix F.

As a solution, we provide single-file reimplementations of a range of existing model-free (BC, TD3-BC, ReBRAC, IQL, SAC-N, LB-SAC, EDAC, CQL, DT) and model-based (MOPO, MOReL, COMBO) methods. Our implementation has a number of advantages. Firstly, we focus on code clarity and minimal code edits between algorithms, leading to a dramatic reduction in code differences between algorithms. Secondly, we implement our algorithms in end-to-end compiled JAX, leading to major speed-ups against competing implementations (Figure 5). We believe these implementations will lead to better algorithm understanding and fairer evaluation, as well as enabling powerful experiments on low compute budgets. We verify the correctness of our reimplementations in Appendix G.

### 4.2 A Unified Hyperparameter Space for Offline RL

Implementation inconsistency and missing ablations are common flaws of offline RL research. The plethora of design decisions in each algorithm obfuscates evaluating how each feature contributes to the performance. To address this, we combine all components from a range of model-free and model-based algorithms (Appendix A) into a unified algorithm and single-file implementation, which we name *Unifloral*. We start by compiling a minimal subspace of components covering the model-free and model-based offline RL algorithms examined in this work (Appendix H). This has a range of hyperparameters in each of four broad design categories, which we identify from prior algorithms: model design, critic objective, actor objective, and dynamics modelling. A more detailed description of design category is in Appendix I.

**Model Design**  The choice of neural network architecture and optimizer is consistent across most offline RL research, with proposed algorithms commonly using multi-layer perceptrons and the Adam optimizer. However, the hyperparameters of these components commonly vary between algorithms. Regarding the model architecture, this includes the number of layers, layer width, and usage of observation and layer normalization. For optimization, this includes the learning rate (shared and actor-specific), learning rate schedule, discount factor, batch size, and Polyak averaging step size. The actor and critic networks can also have different structures, such the size of the critic ensemble and whether the policy stochasticity.

**Critic Objective** The core contribution of offline RL research is often a novel critic objective [12, 13]. However, many of the components in the proposed objectives are shared with prior work. We define the critic objective as the weighted sum of those components, or a selection between them if mutually exclusive, in order to include all referenced methods (except CQL, which we omit due to its substandard performance and high complexity). More detail in Appendix I.1.

**Actor Objective** We define the unified actor loss as the weighted sum of three terms:

$$\mathcal{L}_{\text{actor}} = \beta_q \cdot \mathcal{L}_q + \beta_{\text{BC}} \cdot \mathcal{L}_{\text{BC}} - \beta_{\mathcal{H}} \cdot \mathcal{H}(\pi(\cdot|s_t)). \tag{2}$$

This consists of $q$ loss $\mathcal{L}_q$, behaviour cloning loss $\mathcal{L}_{\text{BC}}$, and policy entropy $\mathcal{H}(\cdot)$, with coefficients $\beta_q, \beta_{\text{BC}}, \beta_{\mathcal{H}} \in \mathbb{R}$ controlling the weight of these terms. More detail in Appendix I.2.

**Dynamics Modelling** We include optional dynamics model training and sampling, broadening Unifloral's coverage to include model-based methods. As is standard, we use an ensemble of dynamics models $\hat{\boldsymbol{D}}_\theta = \{\hat{D}_\theta^1, \hat{D}_\theta^2, ..., \hat{D}_\theta^M\}$, where each $\hat{D}_\theta^i$ is trained to predict state transitions and rewards. Following MOPO, we penalize the agent for going to states where the ensemble disagreement is high as measured by the standard deviation of the model's predictions. More in Appendix I.3.

# 5 Novel Methods Research with Unifloral

Our unified algorithm and hyperparameter space enable researchers to combine different components and search through algorithm designs by only modifying the configuration of the unified implementation. To demonstrate the avenues our work opens up and encourage further research, we provide two "mini-papers" completed entirely by specifying configurations of the unified implementation, without any code changes. We examine a model-free and a model-based improvement.

## 5.1 TD3 with Advantage Weighted Regression

**Hypothesis** In Section 3.3, we show that one of two methods consistently outperformed existing baselines: ReBRAC [5] and IQL [21]. ReBRAC is derived from TD3-BC, meaning it optimizes its actor using TD3 value loss in combination with a BC loss term for regularization. In contrast, IQL uses only a BC loss but performs *advantage weighted regression* (AWR) by weighting the BC loss of each action by its estimated advantage. We hypothesise that substituting the BC term in ReBRAC with AWR, a method we name **TD3-AWR**, would combine the strengths of these methods and lead to improved performance overall.

**Evaluation** We define TD3-AWR in Unifloral by using the AWR hyperparameters from IQL and the ReBRAC hyperparameters elsewhere. In Figure 6, we show that TD3-AWR's performance curve strictly dominates ReBRAC on 6 out of 9 datasets and is dominated by ReBRAC in only 1. Interestingly, TD3-AWR achieves superior performance to ReBRAC under few policy evaluations—such as in halfcheetah-medium-expert and pen-expert—despite searching over a wider range of hyperparameters. Similarly, TD3-AWR strictly dominates IQL on 7 datasets, thereby outperforming both of its source algorithms.

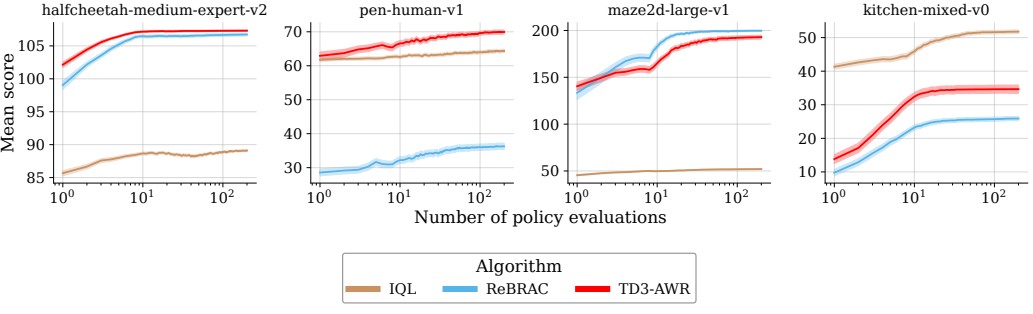

Figure 6: TD3-AWR evaluation against ReBRAC and IQL (full results in Appendix J).

## 5.2 Improving Policy Optimization for Model-Based Offline RL

**Hypothesis** In Section 3.3, we demonstrate the poor performance of model-based methods on non-locomotion environments. Whilst this is partially due to overfit hyperparameters, the design space of policy optimizers in model-based methods is underexplored, with all considered methods using SAC-N or CQL (Figure 5). Given the performance improvements from recent methods, we posit that these methods would be more competitive with an alternative policy optimizer. We therefore propose using ReBRAC with synthetic rollouts generated from a MOPO world model, which we name **Mo**del-based **B**ehaviour **R**egularized **A**ctor-**C**ritic, or **MoBRAC**.

**Evaluation** We implement MoBRAC in Unifloral, using the MOPO hyperparameters for dynamics model training and sampling, then using the ReBRAC hyperparameters elsewhere. Figure 7 shows how MoBRAC outperforms other model-based methods for all datasets, except for MOPO in `maze2d-large-v1`. Under a transparent evaluation budget, we find that MoBRAC outperforms the other model-based methods in 6 out of 9 datasets and is tied with MOPO for 3 others (Appendix K).

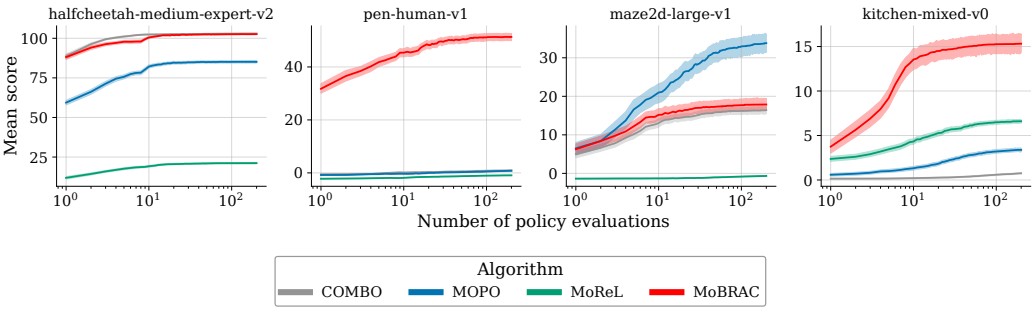

Figure 7: MoBRAC evaluation against prior model-based algorithms (full results in Appendix K).

## 6 Related Work

Our work builds upon several foundational aspects of offline RL, including evaluation strategies, open-source implementations, and algorithmic unification. Existing evaluation regimes primarily address hyperparameter tuning either through limited online interactions [8, 9] or by estimating policy performance offline [4, 10, 22]. In contrast, our approach introduces an evaluation procedure that requires neither reference policies nor additional hyperparameters, offering broader applicability across the entire D4RL benchmark suite. Furthermore, our single-file implementations draw inspiration from projects such as CORL [17, 23] and CleanRL [24], whilst our unified algorithm, Unifloral, is informed by prior unification attempts [25, 26, 15, 27]. For a comprehensive review, see Appendix L.

## 7 Conclusion

In this work, we addressed critical challenges in problem formulation, evaluation, and algorithm unification in offline RL. We introduced a taxonomy that clearly distinguishes between offline RL variants—spanning zero-shot deployment to approaches with limited pre-deployment tuning or post-deployment adaptation. This categorization exposes the hidden online interactions, such as hyperparameter tuning, that have long confounded fair evaluation and reproducibility. To overcome these issues, we proposed a rigorous evaluation procedure that transparently quantifies the cost of online interactions via noisy, single-episode feedback. Additionally, by dissecting components of existing offline RL algorithms, we developed *Unifloral*, a novel unified offline RL algorithm that combines improvements of many previous methods, enabling seamless ablation of algorithmic components. We demonstrate this with two novel algorithms inside Unifloral, TD3-AWR and MoBRAC, which integrate the strengths of existing methods to achieve superior performance over a wide range of tasks. Collectively, our contributions set a new standard for addressing ambiguity in offline RL, promoting rigorous evaluation, and driving reproducible, impactful research in the field.

## Acknowledgements

The authors thank Michael Beukman, Cong Lu, Jack Parker-Holder, Hugh Bishop, and Nathan Monette for their valuable feedback on the paper. MJ, UB, and JL are funded by the EPSRC Centre for Doctoral Training in Autonomous Intelligent Machines and Systems. MJ is also funded by Amazon Web Services, UB is also funded by the Rhodes Scholarship and JL is funded by Sony Interactive Entertainment Europe Ltd.

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

# A  Algorithm Implementations in Unifloral

## A.1  Model-Free Offline RL

**SAC**  Soft Actor-Critic (SAC) by Haarnoja et al. [28] is a $Q$-learning method with a stochastic actor. The authors use two independently optimized $Q$-functions and take their minimum for the value function gradient to reduce positive bias in the policy improvements. SAC uses function approximators for both the policy and value functions.

**EDAC**  Function approximators do not operate well out-of-distribution (OOD), which poses a significant challenge for offline RL methods that rely on a fixed dataset of logged trajectories. An et al. [13] propose increasing the size of the $Q$-function ensemble. They find that SAC requires a large ensemble to avoid optimistic value estimations for OOD actions as the cosine similarity of the gradients increases. To minimize this similarity within the ensemble, the authors propose the Ensemble-Diversified Actor-Critic (EDAC), which adds an ensemble similarity penalty to the $Q$-function loss in SAC. We refer to SAC with more than two members in the ensemble as **SAC-N**.

**CQL**  Optimistic value estimations when bootstrapping from OOD actions is a persisting issue in offline RL. Kumar et al. [12] propose learning a conservative $Q$-function that lower bounds the true value. They perform SAC updates to the $Q$-function with an additional minimization term that uses the value of randomly sampled actions. Their Conservative $Q$-Learning (CQL) algorithm is also implemented on top of a SAC-N policy update similar to EDAC.

**TD3-BC**  Fujimoto et al. [29] formulate the Twin-Delayed Policy Deep Deterministic policy gradient algorithm (TD3) to address the value estimation pathology in *online* RL where the ensemble of $Q$-networks is updated at a higher frequency than the actor. TD3 also takes the minimum over the critics ensemble as in CQL, SAC-N, and EDAC. Follow-up work by Fujimoto and Gu [3] adapts the method for the *offline* paradigm by adding a behaviour cloning (BC) regularization term to the actor's updates. This augmented algorithm is commonly referred to as TD3-BC. Not having to update two networks in every training step brings significant speed-ups while still matching the highest scores across all D4RL [20] locomotion tasks at an increased stability.

**IQL**  Implicit $Q$-learning by Kostrikov et al. [21] is a computationally efficient algorithm that avoids querying out-of-sample actions altogether by using *expectile regression*. The $Q$-function is updated using a mean squared error loss on state-action pairs from the dataset. This approximation of the optimal $Q$-function is used to extract the policy through advantage-weighted regression [30], where each action is weighted according to the exponentiated advantage with an inverse temperature hyperparameter that directs the policy towards higher $Q$-values when increased and approximates behavior cloning [31] when decreased.

**ReBRAC**  Tarasov et al. [5] use the Behavior Regularized Actor-Critic (BRAC) framework [32] and the behavior cloning term from TD3-BC [3] to propose the Revisited BRAC algorithm (ReBRAC). Specifically, they decouple the BC penalty coefficient in the critic and the actor objectives, thus requiring additional hyperameteres to the benefit of higher scores and faster convergence on D4RL. In addition, ReBRAC [5] proposes several improvements, like using deeper networks, training with larger batches, adding layer norms to the critic network, and changing the $\gamma$ hyperparameter for tasks with different reward sparsity. However, these design decisions add new hyperparameters with tuning overheads since they are reportedly different for each D4RL dataset.

## A.2  Model-Based Offline RL

**MOPO**  In Model-Based Offline Policy Optimization (MOPO), Yu et al. [2] argue that offline RL algorithms should be able to go beyond the behaviors in the data manifold to avert sub-optimalities in the dataset and generalize to new tasks to deliver on the promises of real-world deployment. MOPO provides several bounds and theoretical guarantees on behavior policy improvement. The model is implemented through an ensemble of multiple dynamics models trained via maximum likelihood. For every policy step during training, the maximum standard deviation of the learned models' prediction at that step is subtracted from the reward. The highest results are obtained on short truncated rollouts

that are 0.5% to 1% of the real environment's episode length. The model predictions are used to form the batch for the SAC [28] policy update step.

**MOReL**    The model-based offline RL algorithm (MOReL) by Kidambi et al. [11] claims to not require severely truncated rollouts due to learning a pessimistic MDP (P-MDP) that is implemented in a similar way to the MOPO dynamics model with an additional early termination condition in the event of high ensemble disagreement. This scalar halting threshold is calculated by taking the maximum distance between the predictions of any two models of the ensemble for every state and action pair in the dataset. Even for academic demonstration datasets like D4RL, this poses a major overhead in addition to model and policy training. The reported rollout length approximating 50% of the original episode length is only achievable through extensive tuning of the pessimism coefficient that scales the discrepancy threshold.

**COMBO**    Conservative Offline Model-Based Policy Optimization (COMBO) by Yu et al. [14] is implemented on top of MOPO [2] with more policy improvement guarantees. They use a CQL [12] policy update step with an added loss term using transitions from the dataset to penalize $Q$-values on likely out-of-support state-actions while increasing $Q$-values on trustworthy pairs. There are many similarities across model-based methods, and many of their algorithmic contributions like the P-MDP from MOReL, uncertainty penalties from MOPO, and the policy update from COMBO can be combined through our framework.

### A.3   Imitation Learning

This section examines methods that operate outside traditional RL paradigms. These methods use identical offline RL datasets and have achieved scores comparable to other offline RL methods when evaluated under the same conditions.

**BC**    Behavioral cloning (BC), originally formalized by Pomerleau [31], directly optimizes the actor by learning the transitions from the dataset in a supervised manner, thus making the final online performance fully reliant on the quality of the dataset. Recent work by Kurenkov and Kolesnikov [9] further points out the effectiveness of BC under restricted budgets.

**DT**    Introduced by Chen et al. [33], Decision Transformers (DT) have shown remarkable generalization [34] in ORL. DT bypasses the need for traditional RL algorithms to use discounted rewards and bootstrapping for long-term credit assignment by using the logged environment interactions as a sequence modelling objective. Instead of sampling from a policy conditioned on the current states, the trained transformer autoregressively generates the next action based on a fixed intra-episode context of previous interaction and a target cumulative return. This target return can be a hyperparameter that significantly increases the tuning overhead if its value is unknown or a way to obtain optimal performance results when the target return is known.

The reward at each step is decremented from the target return, which is referred to as *return-to-go* at time $t$. Formally, $\hat{R}_t = \sum_{t'=t}^{T} r_{t'}$ where $r_{t'}$ are the observed rewards. Rather than directly modelling the reward function $R$, the model is conditioned on the return-to-go values to enable generation based on desired future returns.

The trajectory representation $\tau$ is structured as an ordered sequence of return-to-go values, states, and actions:
$$\tau = (\hat{R}_1, s_1, a_1, \hat{R}_2, s_2, a_2, ..., \hat{R}_T, s_T, a_T), \tag{3}$$

where $(s_t, a_t) \in \mathcal{S} \times \mathcal{A}$ for all timesteps $t$.

During online evaluation, the model is initialized with a desired target return and an initial state $s_0 \sim \mathcal{S}_0$. After executing action $a_t$, the received reward is subtracted from the target: $\hat{R}_{t+1} = \hat{R}_t - r_t$.

# B  Full Results from the Proposed Evaluation Procedure

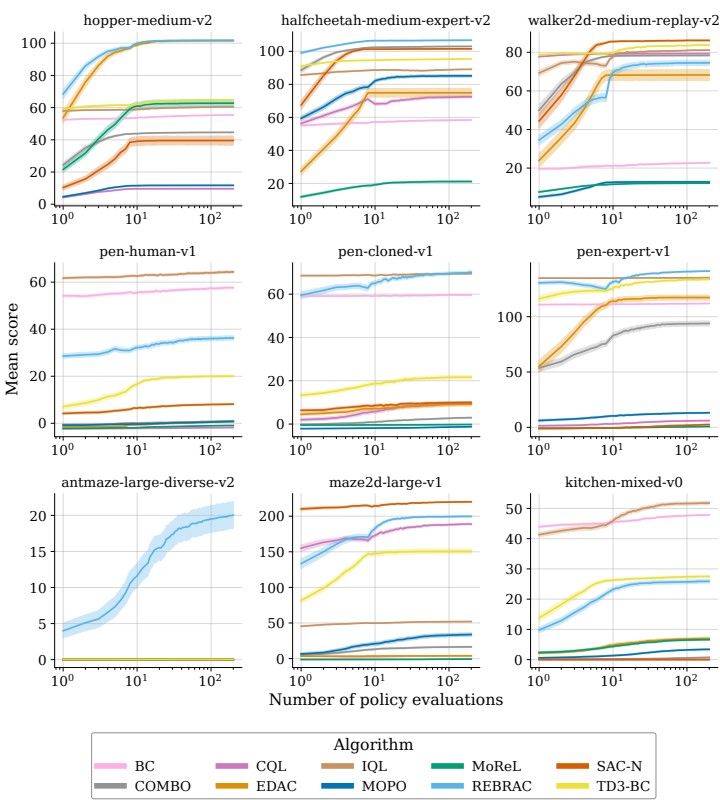

Figure 8: Evaluation of prior algorithms—mean and 95% CI over 500 bandit rollouts, with $K = 8$ policy arms subsampled from 20 trained policies each rollout. The $x$-axis denotes the number of bandit pulls, whilst the $y$-axis denotes the true expected score of the *estimated best arm* after $x$ pulls.

# C Evaluation Benchmarks in Prior Work

Table 1: Evaluations performed in the papers introducing the offline RL algorithms we consider. A "✓" indicates complete evaluation, "∼" indicates a partial evaluation, and "−" indicates that the domain was not evaluated. MuJoCo locomotion is the most widely studied domain, although random and expert datasets are often omitted. Atari experiments are limited to only 5 datasets (Breakout, Qbert, Pong, Seaquest, and Asterix). Notably, the model-based offline RL works referenced here only evaluate on locomotion tasks, which may explain their dramatic performance collapse on non-locomotion tasks.

| Algorithm | D4RL Fu et al. [20] | | | | | | | | Atari |
|---|---|---|---|---|---|---|---|---|---|
| | Locomotion | Adroit | Kitchen | Maze2d | AntMaze | Minigrid | Carla | Flow | |
| CQL [12] | ∼ | ✓ | ✓ | − | ✓ | − | − | − | ∼ |
| DT [33] | ∼ | − | − | − | − | − | − | − | ∼ |
| EDAC [13] | ✓ | ✓ | − | − | − | − | − | − | − |
| IQL [21] | ∼ | ✓ | ✓ | − | ✓ | − | − | − | − |
| ReBRAC [5] | ✓ | ✓ | ✓ | ✓ | ✓ | − | − | − | − |
| SAC-N [13] | ✓ | ✓ | − | − | − | − | − | − | − |
| TD3-BC [3] | ✓ | − | − | − | ∼ | − | − | − | − |
| COMBO [14] | ∼ | − | − | − | − | − | − | − | − |
| MOPO [2] | ∼ | − | − | − | − | − | − | − | − |
| MOReL [11] | ∼ | − | − | − | − | − | − | − | − |

# D Distractor Policy Phenomenon

Here, we show additional observations from the analysis of distractor policies in Section 3.3.

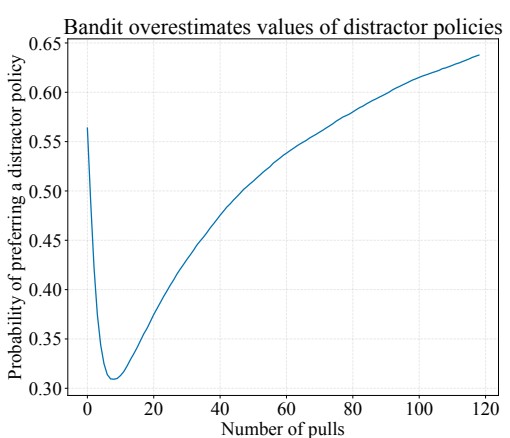

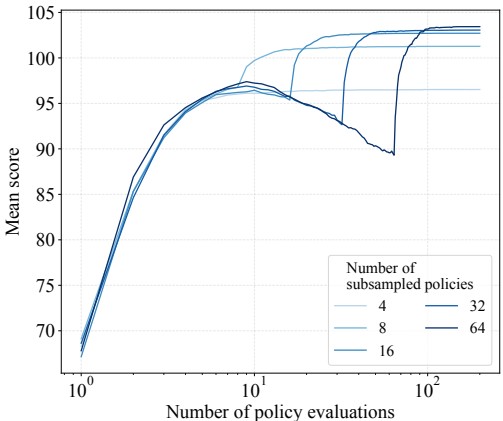

(a) Probability of preferring a distractor policy (inside dashed orange lines in Figure 4) against the number of pulls (mean over 100K random policy orderings). The probability of preferring an unstable policy increases over time.

(b) The number of subsampled policies influences evaluation behaviour—as the number of policies increases, we observe a greater dip in selected-policy performance from our UCB bandit. This is due to the presence of *distractor policies* (Figure 4), which achieve higher peak performance with a lower mean.

# E  Code Philosophy

## E.1  Single-file

We follow the community's preference for single-file algorithm implementations with integrated loggers and evaluations [17, 23, 35, 24]. All of our model-free algorithm implementations are self-contained, with every object necessary to set the hyperparameters, run the training loop, and evaluate the policy included in a single file. As model-based methods typically run sequential dynamics and policy training phases, we implement a single-file dynamics training script that saves trained model checkpoints. These can then be imported by any of the policy training scripts for the model-based algorithms.

## E.2  Consistent

Even within the same library, algorithm implementations often differ in boilerplate code. We change the minimum number of lines between implementations to control for implementation differences and help developers. Guided by the design genealogy illustrated in Figure 10a, we first ensure the single file implementation of the base algorithms like BC and SAC-N is clear and concise (Figure 10b) and then make minimal differences from their algorithmic *ancestors* (Figure 10c).

Figure 11a shows the minimal differences between clean implementations of each algorithm, and Figure 11b shows the line differences from CQL. We acknowledge that prior implementations do not directly seek to minimize the differences between single-file implementations, but we believe it to be a beneficial feature for research. See Figure 12 and Figure 13 for a more complete illustration of the full code.

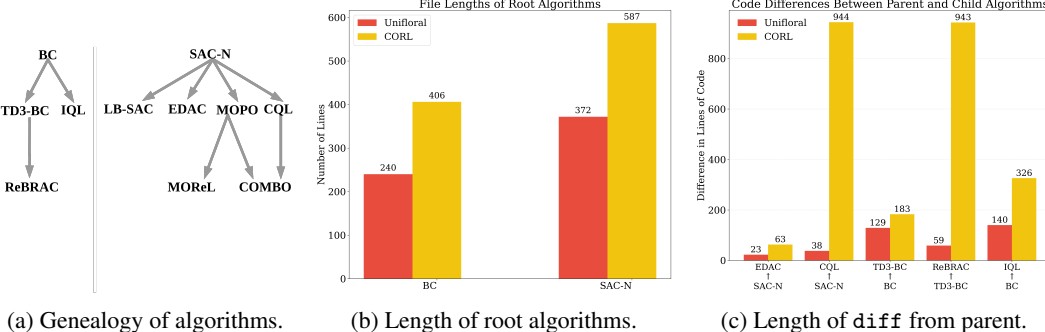

(a) Genealogy of algorithms.   (b) Length of root algorithms.   (c) Length of `diff` from parent.

Figure 10: We provide clean and consistent single-file implementations, as demonstrated by compact implementations and minimal differences between algorithms.

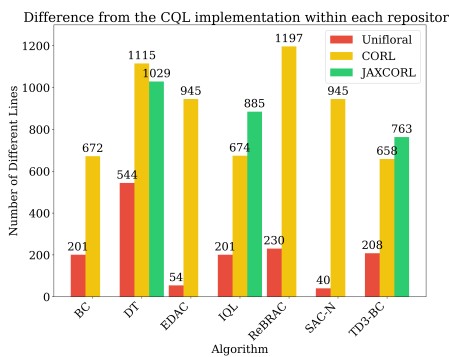

(a) Using command line tool `diff` on our implementations of SAC-N and EDAC.

(b) Implementation length difference of each algorithm from CQL in their respective repository.

Figure 11: Analysis of algorithmic differences between offline RL implementations.

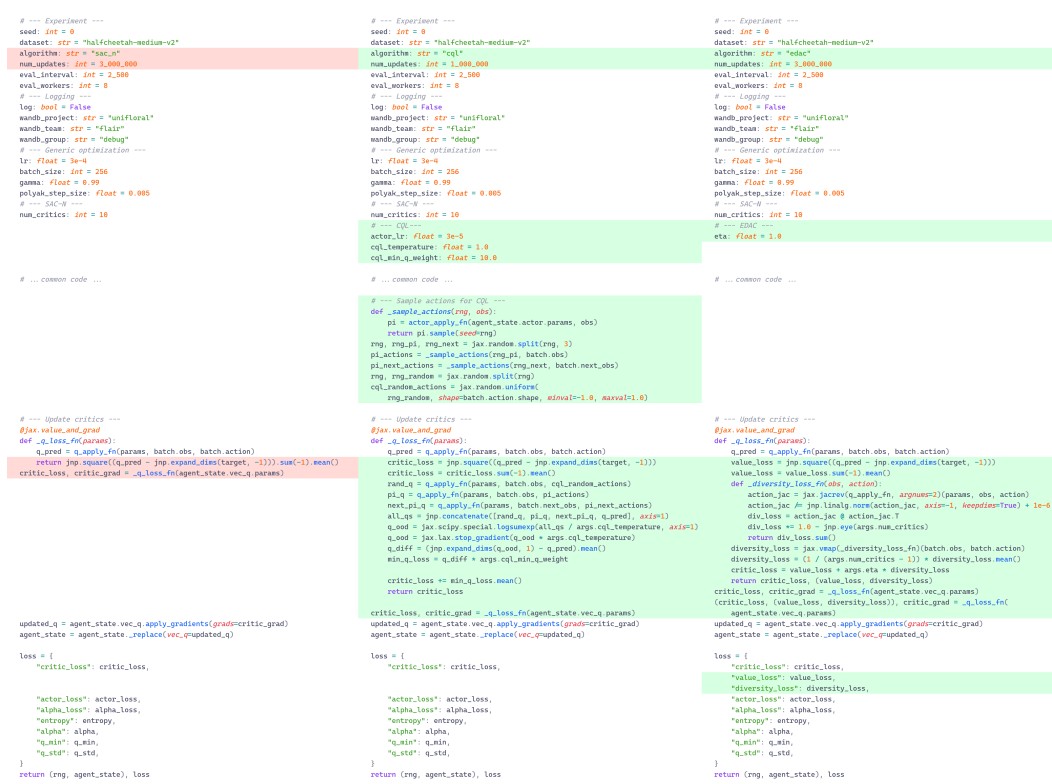

Figure 12: All code edits across implementations, from left to right: SAC-N, CQL, and EDAC.

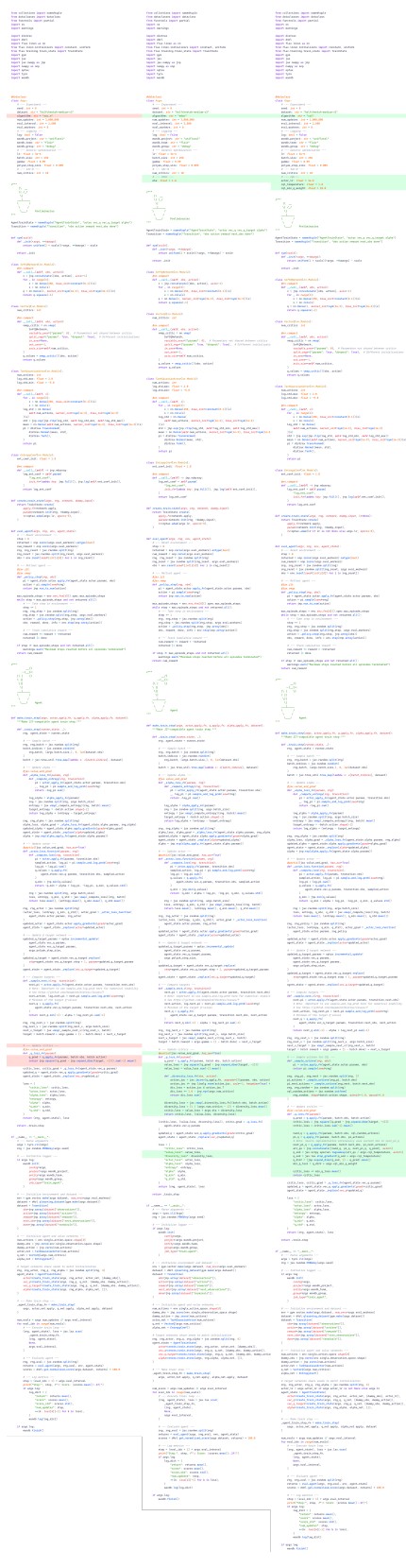

Figure 13: Full code difference for SAC-N, EDAC, and CQL from left to right. The code for the final evaluation loop is omitted to illustrate the consistency of the algorithm implementations.

# F  Reimplementation Training Time

Table 2: Speed up from our JAX implementations, training time in minutes. Algorithms trained for 1M update steps on `HalfCheetah−medium−expert` using a single L40S GPU. Our library, Unifloral, is the fastest across the board.

| Algorithm | OfflineRL-Kit | CORL | JAX-CORL | Unifloral |
|-----------|--------------|------|----------|-----------|
| BC | 19.8 | 15.0 | — | 1.7 |
| TD3-BC | 56.1 | 42.5 | 6.9 | 3.1 |
| IQL | 79.7 | 65.0 | 5.2 | 4.0 |
| ReBRAC | — | 8.7 | — | 6.8 |
| SAC-N | 107.5 | 98.8 | — | 7.7 |
| CQL | 203.9 | 180.3 | 20.7 | 9.8 |
| EDAC | 127.1 | 113.0 | — | 20.8 |
| MOPO | 168.1 | — | — | 14.0 |
| MOReL | — | — | — | 14.0 |
| COMBO | 289.6 | — | — | 22.0 |

# G  Results Reproduction

Table 3: Performance of our algorithm reimplementations over 5 training seeds, Mean±Std.

| Env. | Dataset | BC | COMBO | CQL | EDAC | IQL | MOPO | MOREL | ReBRAC | SAC-N | TD3-BC |
|------|---------|----|-------|-----|------|-----|------|-------|--------|-------|--------|
| HalfCheetah | Expert | 93.0 ± 0.4 | 89.5 ± 9.3 | 3.3 ± 1.3 | 2.3 ± 0.0 | 96.3 ± 0.3 | 62.7 ± 19.1 | 43.0 ± 27.2 | 106.3 ± 0.9 | 98.8 ± 2.8 | 98.0 ± 0.8 |
| | Medium | 42.5 ± 0.2 | 72.2 ± 1.5 | 63.9 ± 1.1 | 52.2 ± 28.0 | 48.5 ± 0.4 | 72.8 ± 0.9 | 72.1 ± 1.6 | 65.6 ± 1.3 | 65.2 ± 1.4 | 48.6 ± 0.3 |
| | Medium-Expert | 59.4 ± 10.9 | 93.6 ± 4.7 | 66.1 ± 8.3 | 102.8 ± 1.1 | 92.3 ± 3.1 | 80.9 ± 19.2 | 63.2 ± 6.8 | 104.5 ± 2.3 | 103.4 ± 5.6 | 92.9 ± 3.5 |
| | Medium-Replay | 37.3 ± 2.0 | 54.4 ± 13.6 | 55.2 ± 1.1 | 55.8 ± 1.0 | 43.8 ± 0.5 | 69.0 ± 1.5 | 65.4 ± 3.5 | 49.1 ± 0.8 | 57.4 ± 1.3 | 44.8 ± 0.5 |
| | Random | 2.2 ± 0.0 | 34.1 ± 1.6 | 30.7 ± 1.1 | 16.8 ± 13.3 | 12.5 ± 3.0 | 30.5 ± 1.0 | 31.8 ± 3.0 | 16.9 ± 17.8 | 26.6 ± 1.0 | 12.0 ± 1.6 |
| Hopper | Expert | 109.5 ± 3.3 | 12.5 ± 15.3 | 1.4 ± 0.3 | 4.9 ± 0.2 | 105.5 ± 4.5 | 2.2 ± 0.8 | 10.6 ± 6.8 | 108.2 ± 4.3 | 93.8 ± 12.2 | 109.4 ± 3.1 |
| | Medium | 55.7 ± 4.8 | 3.1 ± 0.4 | 7.6 ± 0.4 | 100.8 ± 1.7 | 64.7 ± 5.6 | 46.6 ± 51.1 | 27.0 ± 10.4 | 101.8 ± 0.8 | 75.2 ± 36.0 | 62.3 ± 4.9 |
| | Medium-Expert | 53.6 ± 4.4 | 2.8 ± 0.5 | 12.2 ± 3.0 | 109.9 ± 0.3 | 108.4 ± 4.9 | 25.2 ± 47.2 | 77.0 ± 44.4 | 108.0 ± 3.4 | 90.5 ± 22.1 | 105.2 ± 9.3 |
| | Medium-Replay | 25.0 ± 5.3 | 28.1 ± 26.7 | 103.0 ± 0.3 | 101.2 ± 0.4 | 73.5 ± 7.5 | 86.3 ± 28.4 | 47.4 ± 13.8 | 84.4 ± 26.8 | 101.9 ± 0.4 | 51.1 ± 24.0 |
| | Random | 4.9 ± 4.8 | 27.0 ± 8.6 | 22.0 ± 12.8 | 22.6 ± 15.2 | 7.3 ± 0.1 | 31.4 ± 0.0 | 21.9 ± 13.0 | 7.8 ± 1.2 | 26.6 ± 10.5 | 8.4 ± 0.7 |
| Walker2d | Expert | 108.5 ± 0.2 | 22.6 ± 24.0 | 2.4 ± 2.4 | 79.0 ± 45.3 | 112.7 ± 0.5 | 55.5 ± 10.7 | 19.4 ± 21.3 | 112.4 ± 0.1 | 3.2 ± 2.2 | 110.3 ± 0.3 |
| | Medium | 63.8 ± 9.8 | 84.5 ± 0.4 | 87.9 ± 0.6 | 75.1 ± 40.9 | 84.0 ± 2.0 | 81.3 ± 2.6 | 16.4 ± 36.9 | 84.3 ± 2.3 | 87.9 ± 0.6 | 84.5 ± 0.7 |
| | Medium-Expert | 108.1 ± 0.4 | 101.2 ± 0.9 | 88.9 ± 36.3 | 112.9 ± 0.7 | 111.8 ± 0.3 | 110.0 ± 1.5 | 21.7 ± 48.8 | 111.6 ± 0.5 | 114.8 ± 0.7 | 110.1 ± 0.5 |
| | Medium-Replay | 23.8 ± 11.3 | 76.5 ± 2.0 | 79.1 ± 1.6 | 86.9 ± 1.5 | 82.8 ± 3.9 | 11.7 ± 3.3 | -0.2 ± 0.0 | 82.7 ± 5.3 | 82.3 ± 1.6 | 78.4 ± 4.0 |
| | Random | 0.9 ± 0.4 | 3.4 ± 2.6 | 9.1 ± 4.9 | 2.0 ± 0.0 | 4.4 ± 0.8 | 4.3 ± 6.3 | 0.3 ± 0.3 | 17.8 ± 8.9 | 20.7 ± 1.2 | 0.3 ± 0.4 |

Table 3 presents the results achieved by our method reimplementations on locomotion datasets, matching the performance of prior implementations [17]. For our library's completeness, we also implement the Decision Transformer (DT) [33] using the hyperparameters from CORL [17].

Table 4: Performance of our Decision Transformer implementation over 5 training seeds, Mean±Std.

| Env. | Dataset | Decision Transformer |
|------|---------|---------------------|
| HalfCheetah | Expert | 92.9 ± 0.1 |
| | Medium | 42.8 ± 0.5 |
| | Medium-Expert | 92.5 ± 0.2 |
| | Medium-Replay | 37.8 ± 1.3 |
| Hopper | Expert | 110.2 ± 1.5 |
| | Medium | 61.3 ± 5.4 |
| | Medium-Expert | 111.3 ± 0.5 |
| | Medium-Replay | 25.5 ± 8.8 |
| Walker2d | Expert | 108.4 ± 0.2 |
| | Medium | 71.4 ± 6.03 |
| | Medium-Expert | 108.1 ± 0.3 |
| | Medium-Replay | 53.24 ± 13.7 |

# H Unifloral Hyperparameters

Table 5: Hyperparameters of prior algorithms in Unifloral—light gray values indicate inactive settings.

| Hyperparameter | IQL | SAC-N | EDAC | TD3-BC | ReBRAC |
|---|---|---|---|---|---|
| Batch size | 256 | 256 | 256 | 256 | 1024 |
| Actor learning rate | 3e-4 | 3e-4 | 3e-4 | 3e-4 | 1e-3 |
| Critic learning rate | 3e-4 | 3e-4 | 3e-4 | 3e-4 | 1e-3 |
| Learning rate schedule | cosine | constant | constant | constant | constant |
| Discount factor $\gamma$ | 0.99 | 0.99 | 0.99 | 0.99 | 0.99 |
| Polyak step size | 0.005 | 0.005 | 0.005 | 0.005 | 0.005 |
| Normalize observations | True | False | False | True | False |
| Actor layers | 2 | 3 | 3 | 2 | 3 |
| Actor hidden size | 256 | 256 | 256 | 256 | 256 |
| Actor layer normalization | False | False | False | False | True |
| Deterministic policy | False | False | False | True | True |
| Deterministic eval | True | False | False | False | False |
| Apply tanh to mean | True | False | False | True | True |
| Learn action std | True | False | False | False | False |
| Log std min | -20.0 | -5.0 | -5.0 | -5.0 | -5.0 |
| Log std max | 2.0 | 2.0 | 2.0 | 2.0 | 2.0 |
| # of critics | 2 | [5–200] | [10–50] | 2 | 2 |
| Critic layers | 2 | 3 | 3 | 2 | 3 |
| Critic hidden size | 256 | 256 | 256 | 256 | 256 |
| Critic layer normalization | False | False | False | False | True |
| Actor BC coefficient | 1.0 | 0.0 | 0.0 | 1.0 | [5e-4–1.0] |
| Actor Q coefficient | 0.0 | 1.0 | 1.0 | [1.0–4.0] | 1.0 |
| Use Q target in actor | False | False | False | False | False |
| Normalize Q loss | False | False | False | True | True |
| Q aggregation method | min | min | min | first | min |
| Use AWR | True | False | False | False | False |
| AWR temperature | [0.5–10.0] | 1.0 | 1.0 | 1.0 | 1.0 |
| AWR advantage clip | 100.0 | 100.0 | 100.0 | 100.0 | 100.0 |
| Critic BC coefficient | 0.0 | 0.0 | 0.0 | 0.0 | [0–0.1] |
| # of critic updates per step | 1 | 1 | 1 | 2 | 2 |
| Diversity coefficient | 0.0 | 0.0 | [0.0–1e3] | 0.0 | 0.0 |
| Policy noise | 0.0 | 0.0 | 0.0 | 0.2 | 0.2 |
| Noise clip | 0.0 | 0.0 | 0.0 | 0.5 | 0.5 |
| Use target actor | False | False | False | True | True |
| Use entropy loss | False | True | True | False | False |
| Actor entropy coefficient | 0.0 | 1.0 | 1.0 | 0.0 | 0.0 |
| Critic entropy coefficient | 0.0 | 1.0 | 1.0 | 0.0 | 0.0 |
| Use value target | False | False | False | False | False |
| Value expectile | [0.5–0.9] | 0.8 | 0.8 | 0.8 | 0.8 |

# I Unified Algorithm Details

In this section we write out the different design decisions in a unified notation.

## I.1 Critic Objective

First, we compute the value target using one of two methods, selectable via the method configuration:

$$v_{t+1} = \begin{cases} v(s_{t+1}) \\ \min_{n=1}^{N} q'_n(s_{t+1}, \text{clip}(\hat{a}_{t+1} + \text{clip}(\epsilon, \epsilon_{\min}, \epsilon_{\max}), a_{\min}, a_{\max})) \end{cases}, \tag{4}$$

where $N$ is the number of ensemble members, $v$ is a value function trained with expectile regression (as in IQL [21]), $\hat{a}_{t+1} \sim \pi(a_{t+1}|s_{t+1})$ is an action sampled from $\pi$ (or a Polyak averaged target policy), $\epsilon \sim \mathcal{N}(0, \sigma^2)$ is random action noise with standard deviation $\sigma$, and $\epsilon_{\min}, \epsilon_{\max}, a_{\min}$, and $a_{\max}$ are clipping ranges. The value target is then augmented with behaviour cloning and entropy terms (coefficients $\alpha_{\text{BC}}$ and $\alpha_{\mathcal{H}}$), defined as

$$\hat{v}_{t+1} = v_{t+1} + \alpha_{\text{BC}} \cdot (\tilde{a}_{t+1} - a_{t+1}) + \alpha_{\mathcal{H}} \cdot \mathcal{H}(\pi(\cdot|s_{t+1})), \tag{5}$$

which is then used to compute the value loss,

$$\mathcal{L}_v = \sum_{n=1}^{N} (q_n(s_t, a_t) - (r + (1-d) \cdot \gamma \cdot \hat{v}_{t+1}))^2. \tag{6}$$

Finally, we add the critic diversity loss term from EDAC [13] with coefficient $\alpha_{\text{div}}$, giving the final critic loss

$$\mathcal{L}_{\text{critic}} = \mathcal{L}_v + \frac{\alpha_{\text{div}}}{N-1} \cdot \sum_{1 \leq i \neq j \leq N} \langle \nabla_{a_t} q_i(s_t, a_t), \nabla_{a_t} q_j(s_t, a_t) \rangle. \tag{7}$$

## I.2 Actor Objective

We write the actor loss as:

$$\mathcal{L}_{\text{actor}} = \beta_q \cdot \mathcal{L}_q + \beta_{\text{BC}} \cdot \mathcal{L}_{\text{BC}} - \beta_{\mathcal{H}} \cdot \mathcal{H}(\pi(\cdot|s_t)). \tag{8}$$

This consists of $q$ loss $\mathcal{L}_q$, behaviour cloning loss $\mathcal{L}_{\text{BC}}$, and policy entropy $\mathcal{H}(\cdot)$, with coefficients $\beta_q, \beta_{\text{BC}}, \beta_{\mathcal{H}} \in \mathbb{R}$ controlling the weight of these terms.

The first term, $\mathcal{L}_q$ is defined simply by a selectable aggregation function over the $q$-network ensemble, with the minimum being the most common choice,

$$\mathcal{L}_q = \begin{cases} -\min_{n=1}^{N}(q_n(s_t, a_t)) \\ -\frac{1}{N} \sum_{n=1}^{N} q_n(s_t, a_t) \\ -q_0(s_t, a_t) \end{cases}. \tag{9}$$

This term may also be normalized across the batch in order to stabilize learning. The second term, $\mathcal{L}_{\text{BC}}$, is most commonly defined as the distance $d$ between the target policy and dataset action, being the mean squared error for deterministic policies or log-probability for stochastic policies. However, some methods use *advantage weighted regularization* (AWR), which further weights this loss by the clipped and exponentiated advantage of the behaviour policy action in order to clone only positive actions within the dataset. Therefore, this term has the following variants:

$$d = \begin{cases} (a_t - \hat{a}_t)^2 \\ -\log \pi(a_t|s_t) \end{cases}, \qquad \mathcal{L}_{\text{BC}} = \begin{cases} d \\ d \cdot \min\left(A_{\max}, e^{\eta \cdot (q(s_t, a_t) - V(s_t))}\right) \end{cases}, \tag{10}$$

where $\eta$ and $A_{\max}$ are the temperature and maximum value for exponential advantage.

## I.3  Dynamics Modelling

We use an ensemble of dynamics models $\hat{\boldsymbol{D}}_\theta = \{\hat{D}_\theta^1, \hat{D}_\theta^2, ..., \hat{D}_\theta^M\}$, where each $\hat{D}_\theta^i$ is trained to predict state transitions and rewards. Following MOPO, we penalize the agent for going to states where the ensemble disagreement is higher, as measured by the standard deviation of the model's predictions. This is used to penalize the reward during policy optimization with a pessimism coefficient $\eta$,

$$\hat{R}(s_t, a_t) = \frac{1}{M} \sum_{m=1}^{M} R_\theta^m(s_t, a_t) - \eta \cdot \sigma(\hat{\boldsymbol{D}}_\theta^{\Delta s}(s_t, a_t)), \tag{11}$$

where $\sigma(\hat{\boldsymbol{D}}_\theta^{\Delta s}(s_t, a_t))$ represents the standard deviation across the models' *state-change* predictions and $R_\theta^m(s_t, a_t)$ is reward prediction of the $m$-th ensemble member.

## J  Complete TD3-AWR Results

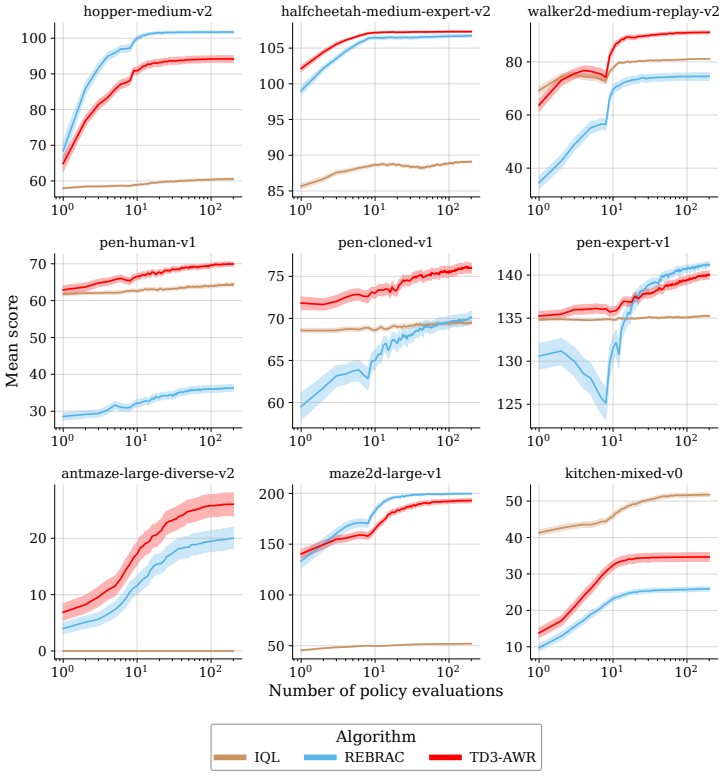

Figure 14: Full comparison of TD3-AWR to prior model-based methods across all datasets.

# K   Complete MoBRAC Results

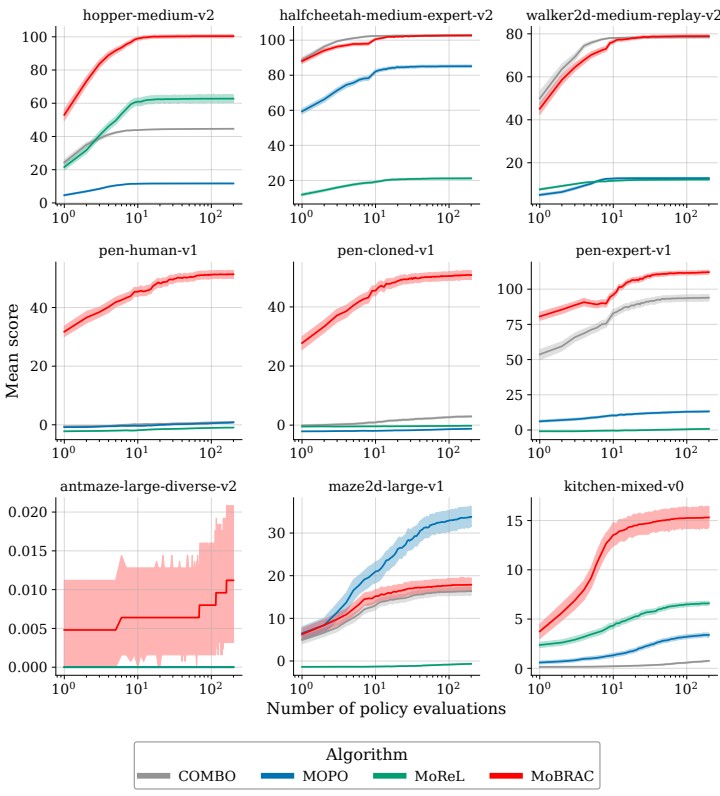

Figure 15: Full comparison of MoBRAC to prior model-based methods across all datasets.

# L Full Related Work

In this section, we describe the prior work related to our evaluation procedure, implementation, and unified algorithm. We implement a comprehensive selection of offline RL algorithms, for which more information can be found in Appendix A.

## L.1 Evaluation Regimes for Offline RL

The challenge of hyperparameter tuning in RL spans various domains. Wang et al. [36] discuss offline tuning and the practical risks of deploying policies of unknown quality in the real world, whilst Paine et al. [4] directly tackle this issue, estimating the *zero-shot* performance of offline-trained policies without any prior online interactions. Their evaluation is limited to behavioural cloning [31, BC] and two critic-based methods, which have since been outperformed by modern algorithms. Konyushova et al. [7] extend this procedure with an online phase, using a UCB-based bandit to investigate policy selection over multiple online evaluations. Further highlighting these challenges, Smith et al. [10] propose a procedure where offline evaluation methods are first calibrated using policies of known quality, evaluating on D4RL [20] locomotion tasks. Unlike their work, we evaluate across the D4RL suite and introduce a procedure that eliminates the need for reference policies or additional hyperparameters. Matsushima et al. [8] present a variant of offline RL that uses a limited number of *online* deployments to update the dataset and iteratively train offline to match the performance of online methods, introducing an online deployment frequency hyperparameter. Kurenkov and Kolesnikov [9] address the practice of unreported online evaluations for hyperparameter tuning, demonstrating how the performance of each algorithm changes with the number of online evaluations. Unlike our procedure, they assume a low-variance estimate of a policy's true performance each evaluation but still conclude that BC outperforms all baselines.

## L.2 Open-Source Implementations

**Offline RL**  Inspiring our implementation, Clean Offline RL [17, CORL] provides single-file implementations of model-free offline RL methods in PyTorch. JAX-CORL [23] is a JAX-based port of CORL, albeit with a limited range of only model-free algorithms, slower training time than our implementations, and lacking our evaluation procedure and code consistency. OfflineRLKit [16] and d3rlpy [37] implement a range of offline RL methods and feature both model-based and model-free methods. Although the repository has transparent class inheritance and polymorphism, it lacks any further attempt at algorithmic unification.

**Online RL**  StableBaselines3 [38] is a set of reliable RL algorithm implementations in PyTorch with the aim of abstracting away training and deployment through an object-oriented interface. SpinningUp [39] is a similar, education-oriented effort of jointly implementing various online RL algorithms. CleanRL [24] follows a different design philosophy with method-focused, single-file implementations of online RL algorithms in PyTorch and JAX. PureJaxRL [35] also follows the single-file approach and is implemented in JAX. Rejax [40] is a popular multi-file JAX-based implementation of PureJaxRL with extensive logging integration and a selection of SOTA methods.

CleanRL, CORL, and JAX-CORL provide clear and accessible logs of their final runs, a standard of reproducibility we plan to uphold throughout every release of our work.

## L.3 Method Unification

Our unified algorithm, Unifloral, is heavily inspired by prior work that also seeks to ablate and unify a range of methods. Lu et al. [15] investigate the key components of model-based offline RL algorithms to find an optimized algorithm that outperforms all model-based baselines. Sikchi et al. [27] cast multiple offline RL methods in the same dual optimization framework and use this unification to categorize them in regularized policy learning and pessimistic value learning. Prudencio et al. [6] provide a survey of offline RL, focused on elucidating the taxonomy and disambiguating the contributions of each algorithm. In online RL, Hessel et al. [25] combine independent components of Deep $Q$-network algorithms into a unified algorithm, Rainbow, reaching SOTA in the Atari 2600 benchmark. Muesli [26] examines the combination of policy optimization and model-based methods.

