# OpenReview forum: "A Clean Slate for Offline Reinforcement Learning"
_NeurIPS.cc/2025/Conference — NeurIPS 2025 oral_

### Official Review · Reviewer_eMD6 · 2025-06-22

**Clarity:** 2
**Significance:** 3
**Originality:** 2
**Rating:** 5
**Confidence:** 4

**Summary:**

The paper identifies a key challenge in offline RL research, the reliance on online evaluations and hyperparameter tuning, which undermines the premise of offline learning. The authors argue that many state-of-the-art algorithms function as black boxes due to their opaque design. To address this, they provide clean, single-file implementations of several well-known offline RL algorithms, offering significant speed-ups over existing libraries. Additionally, they propose two novel algorithmic paradigms—TD3-AWR and MoBRAC—and provide extensive empirical evaluations.

**Questions:**

1. The authors attribute speed-ups relative to CORL to their use of JAX. Could they elaborate on why their implementation outperforms JAX-CORL? Also, could they provide a comparison with d3rlpy, another widely used offline RL library?
2. In Taxonomy point 2b, if only one policy is eventually deployed, how is the selection carried out based solely on evaluation?
3. Which hyperparameter selections are the authors targetting through online finetuning?  What is the range of these hyperparamters? Since hyperparameters often have continuous domains, this could result in a prohibitively large search space. A table of algorithms with hyperparameters, ranges and online finetuning steps and best hyperparameter found will be helpful for the community.
4. How many iterations does the bandit algorithm require for convergence in practice? Can convergence be quantified or benchmarked?
5. Why is the Decision Transformer (DT) not included in Figure 3? Also how does MoBRAC handle the overfiiting observed by model based RL since it uses MOPO to learn the world dynamics.
6. Why does MoBRAC exhibit such high variance in the AntMaze task (Fig. 15)? Is this due to instability in model-based rollouts or something else?
7. How sensitive are the results to random seeds or initialization?

Please also refer to the weaknesses.

**Ethical Concerns:**

["NO or VERY MINOR ethics concerns only"]

**Final Justification:**

A clean offline RL library is useful for the community. The authors addressed some of my concerns in the rebuttal.

**Limitations:**

Yes discussed.

**Paper Formatting Concerns:**

None.

**Quality:**

3

**Strengths And Weaknesses:**

Weaknesses


1. The evaluation is limited to primarily robotic benchmarks. It would be beneficial to include results on additional datasets like those in neorl/neorl2  or other diverse domains [4].
2. The taxonomy omits a class of techniques that tune offline RL algorithms based on Q-values without requiring online interaction. Please consider referencing [1].
3. Minor: The notation in Equation 1 is slightly inconsistent. T is both used a transition probability and as a time subscript in Eq 1. Also in Eq 1 s1:H∼H should this be s1:H∼T as states are generated from transition dynamics?
4. Several recent offline RL methods, such as Sparse Q-Learning (SQL) and Exponential Q-Learning (EQL) [2], as well as A2PR [3], are not discussed or compared.


Strengths


1. The provision of clean implementations and a unified benchmarking framework is an impactful contribution to the offline RL community, promoting transparency and reproducibility.
2. The empirical evaluations are thorough and compelling. I particularly appreciated the finding that “no algorithm consistently performs well across all datasets,” and the revelation of overfitting in model-based algorithms such as MOPO, MOReL, and COMBO.
3. Two new algorithm designs TD3-AWR and MoBRAC—that leverage the unified framework to demonstrate superior performance across various benchmarks.


[1] Kumar, Aviral, et al. "A workflow for offline model-free robotic reinforcement learning." arXiv preprint arXiv:2109.10813 (2021).
[2] Xu, Haoran, et al. "Offline rl with no ood actions: In-sample learning via implicit value regularization." arXiv preprint arXiv:2303.15810 (2023).
[3] Liu, Tenglong, et al. "Adaptive advantage-guided policy regularization for offline reinforcement learning." arXiv preprint arXiv:2405.19909 (2024).
[4] Additional Benchmarks : https://github.com/polixir/NeoRL , https://github.com/polixir/NeoRL2, RL Unplugged: A Suite of Benchmarks for Offline Reinforcement Learning

---

> ### Author Rebuttal · Authors · 2025-07-30
>
> Thank you for your recognition of our extensive empirical studies and constructive criticism of our work.
>
> ## Weaknesses
>
> > It would be beneficial to include results on additional datasets like those in neorl/neorl2 or other diverse domains.
>
> We believe that our robotics benchmarks already capture significant diversity: Across 7 environments and 21 datasets, we include locomotion, manipulation and maze solving. We also evaluate our methods on *all datasets* of the three most common mujoco environments.
> We also agree that neorl2 is an interesting benchmark and are currently looking into adding it for the camera-ready version.
>
> > The taxonomy omits a class of techniques that tune offline RL algorithms based on Q-values without requiring online interaction.
>
> These methods fall under our taxonomie's category (1), since (as the reviewer noted) they use no online interaction before deployment. We will cite the reviewer's suggested reference in the next iteration of our paper.
> Furthermore, we already cite two references that estimate policy performance offline (line 317) and elaborate in more detail in section L.1 of the Extended Related Work in the Appendix.
>
> > Minor: The notation in Equation 1 is slightly inconsistent.
>
> We thank the reviewer for their keen eye and have fixed this inconsistency.
>
> > Several recent offline RL methods, such as Sparse Q-Learning (SQL) and Exponential Q-Learning (EQL) [2], as well as A2PR [3], are not discussed or compared.
>
> We are citing 21 RL methods and implementing 10 algorithms as well as 2 novel algorithms. This includes all offline-only algorithms in (Jax-)CORL and almost all algorithms in OfflineRLKit. We aimed to provide one of the most comprehensive offline RL libraries that currently exist, and argue that we have succeeded.
> Still, we are open to implementing the suggested algorithms and are planning to add them for the camera-ready version. Our work is also intended to be a scaffolding for the community to seemlessly add concise implementation of different methods.
>
> ## Answers to Questions
>
> > Could \[the authors\] elaborate on why their implementation outperforms JAX-CORL?
>
> Profiling the runtime of jax-corl was outside the scope of our work. From looking at their codebase, it seems like they are not using `lax`-based (GPU-compilable) control flow, which can lead to significant performance decreases
>
> > In Taxonomy point 2b, if only one policy is eventually deployed, how is the selection carried out based solely on evaluation?
>
> This depends on the specific offline RL method. For example, Active Offline Policy Selection (Konyushkova et al., 2022, [https://arxiv.org/abs/2106.10251XiV](https://arxiv.org/abs/2106.10251)) uses Bayesian optimization over a Gaussian process of policy values. We elaborate in Appendix L.1 (line 960).
>
> > Which hyperparameter selections are the authors targetting through online finetuning? What is the range of these hyperparamters?
>
> When selecting policies via a bandit, we select between policies trained with different hyperparameter settings. We use the hyperparameter ranges from the respective methods' original papers, as described in line 175.
> Our algorithm's hyperparameter ranges can be found in Appendix H.
>
> > How many iterations does the bandit algorithm require for convergence in practice? Can convergence be quantified or benchmarked?
>
> This is highly dependent on the HP robustness of the evaluated algorithm. For example, if the algorithm produces high-performing policies for a wide range of hyperparameter settings, convergence is fast.
>
> > Why is the Decision Transformer (DT) not included in Figure 3?
>
> We include it in the camera-ready version and present results below. We did not include it in the initial set of algorithms as the Decision Transformer is not a traditional RL algorithm, even though it is technically trained offline on the same datasets. Furthermore, as done in CORL [1] and the original DT paper [2], the sequential nature of the evaluation function is different from the other algorithms. We will report the results from the same DT evaluation setup used by CORL [1].
>
> #### Preliminary DT results
>
> We ran our implementation of the DT algorithm for three seeds each.
>
> | Dataset | Mean | Stdev |
> | --- | --- | ---  |
> | Hopper-medium-v2  | 5.5 (unstable, max 59.6) | 16.3 |
> | Hopper-medium-expert-v2  | 10.9 (unstable, max 111.05) | 33.1 |
> | Halfcheetah-medium-v2  | 42.4 | 0.5 |
> | Halfcheetah-medium-expert-v2  | 74.6 | 23.1 |
> | Walker2d-medium-expert-v2  | 83.4 | 15.1 |
>
>
> > Also how does MoBRAC handle the overfiiting observed by model based RL since it uses MOPO to learn the world dynamics.
>
> MoBRAC uses the *trained* MOPO models as they are. We propose the algorithm as an improvement over current approaches and demonstrate it by intentionally using the same models. MoBRAC is an ablation in itself, showcasing that behavior regularization leads to agents that are more robust, despite being trained using samples from somewhat flawed dynamics models.
>
> > Why does MoBRAC exhibit such high variance in the AntMaze task (Fig. 15)?
>
> The variance is very low in practice, but might appear high due to the extremely small scale of the y axis. Most likely, a few policies have recieved a reward by sheer luck, hence the performance is ever-so-slightly above zero.
>
> > How sensitive are the results to random seeds or initialization?
>
> We did not explicitly investigate this. However, every hyperparameter sample also has a different randomly sampled seed, so our metric evaluates an even stronger form of robustness (which includes hyperparameters).
>
> [1] Tarasov et al. *CORL: Research-oriented Deep Offline Reinforcement Learning Library* (2022)
>
> [2] Chet et al. *Decision Transformer: Reinforcement Learning via Sequence Modeling* (2021)
>
> Thank you for your valuable feedback. Given our responses and clarifications to your questions and weaknesses, we kindly ask you to consider increasing your score.

---

> > ### Comment · Reviewer_eMD6 · 2025-08-03
> >
> > I thank the author for their response and DT experiments. I still feel taxonomy 2b is unclear " Train a set of policies offline, then deploy one of them. After deployment, select which policy to use based on online performance." . I think either a set of policy is deployed rather than one and one of them is finally chosen based on maximum return. I also feel an offline RL library should be comprehensive in covering all types of SOTA algorithms, and would highly recommend the authors to include algos mentioned in point 4 in their implementation. It would also be nice to see some quantification for Q4 in revision.

---

> > > ### Author Response · Authors · 2025-08-03
> > >
> > > Thank you for engaging with our rebuttal! We respond to your remaining questions below:
> > >
> > > **Taxonomy 2b** - We understand how our original wording could be ambiguous, so have updated the wording accordingly:
> > >
> > > > Train a set of policies offline then deploy online, adaptively selecting a policy every episode based on online performance.
> > >
> > > We believe this clearly describes the setting, where a collection of policies are trained on offline data, then a policy selection bandit continually selects policies to maximise cumulative return after deployment. We have updated the manuscript to reflect this change.
> > >
> > > **Addition of methods [2] and [3]** - As stated in our rebuttal, our library and evaluation encompasses almost all algorithms found in the two most popular offline RL libraries, in addition to 2 novel algorithms, making it one of the most comprehensive offline RL libraries and evaluations in existence. Hundreds of offline RL methods have been proposed in recent years, and many of their evaluations have been fundamentally flawed (as discussed in our work), making it impossible to determine the state-of-the-art or provide complete coverage thereof. Whilst we will endeavour to support the proposed algorithms, we do not believe a lack of coverage to be a valid criticism of our work.
> > >
> > > **Summary** - Thank you again for your thoughtful response. Please let us know if you have any remaining concerns regarding the working of our taxonomy or our evaluation's coverage. If not, would you consider increasing your score to reflect these points?

---

> > > > ### Comment · Reviewer_eMD6 · 2025-08-04
> > > >
> > > > Thank you for your response I had already increased my score in the previous iteration.

---

### Official Review · Reviewer_7aG4 · 2025-07-02

**Clarity:** 3
**Significance:** 4
**Originality:** 2
**Rating:** 5
**Confidence:** 4

**Summary:**

This paper tackles two core challenges in offline RL: ambiguous problem definitions and opaque algorithmic implementations. The authors propose a rigorous taxonomy of offline-RL variants and a transparent evaluation procedure that quantifies online tuning budgets via a simulated bandit over collected policy scores. Second, they deliver minimal, single-file JAX implementations of a broad spectrum of model-free and model-based offline-RL methods. Building on these foundations, they introduce a unified framework named Unifloral that unify the hyperparameter space across different algorithms and provide modular implementation of existing Offline-RL methods. Based on this framework, the authors instantiate two new algorithms, TD3-AWR (model-free) and MoBRAC (model-based), and achieve great performance.

**Questions:**

I don't quite understand the difference between pre-deployment and post-deployment Offline-RL (2a and 2b). They both have online evaluations and then policy selection, right? If so, what is the difference? It would be great to have examples to in the supplementary material to give more detailed explanation.

**Ethical Concerns:**

["NO or VERY MINOR ethics concerns only"]

**Final Justification:**

I'm generally satisfied with the paper. The discussion of pre-deployment interaction and post-deployment adaptation in response to reviewer rtSN is comprehensive. While the authors did not include a hyperparameter landscape analysis, I accept their explanation that this is not a standard practice in the field. That said, I believe a more comprehensive analysis of hyperparameter impacts could substantially strengthen the work, as also noted by other reviewers. Overall, I maintain my original score.

**Limitations:**

Yes

**Quality:**

4

**Strengths And Weaknesses:**

Strengths:
 - The explicit taxonomy and simulated-bandit evlaulation expose hidden online tuning costs, enabling fair comparisons across Offline-RL algorithms
 - The single-file JAX reimplementations alleviate the discrepancies among algorithms, providing a unified way for follow-up researchers to perform fair and meaningful baseline comparisons.
 - The novel algorithms, TD3-AWR and MoBRAC show significant gains in model-free and model-based settings.

Weakness:
 - The impact of individual Unifloral components (e.g., different critic objective terms, ensemble sizes, entropy weights) on algorithm performancne is not clear in this paper. I suggest the authors make systematical analysis on these components which may bring more insights.
 - It lacks analysis of the hyperparameter space itself. It would be great to have visualizations of the hyperparameter landscape, e.g., sensitivity heatmaps to show the component interactions.

---

> ### Author Rebuttal · Authors · 2025-07-30
>
> We thank the reviewer appreciating both our novel algorithms, the implementation and taxonomy.
>
> ## Weaknesses
>
> > The impact of individual Unifloral components (e.g., different critic objective terms, ensemble sizes, entropy weights) on algorithm performancne is not clear in this paper. I suggest the authors make systematical analysis on these components which may bring more insights.
>
> We believe that the strong empirical performance of our algorithms demonstrates a clear benefit of combining components that have been independently proposed. Additionally, the algorithms Unifloral is based on represent ablations of Unifloral itself. For example, ReBRAC is an ablation of TD3-AWR (no Adwantage Weighted Regression), and IQL is an ablation of TD3-AWR (basic BC loss instead of TD3-BC value loss). Therefore figure 6 and 7 explicitly analyse the impact of individual Unifloral components.
>
> > It lacks analysis of the hyperparameter space itself. It would be great to have visualizations of the hyperparameter landscape, e.g., sensitivity heatmaps to show the component interactions.
>
> We decided to focus our limited experiment budget on a more comprehensive and complete comparison of methods. Analyses of hyperparameter landscapes are not standard in the field, and we are not aware that any paper we cite or implement visualises sensitivity heatmaps. Therefore, we respectfully disagree that this is a weakness of the paper.
>
>
>
> ## Questions
>
> > I don't quite understand the difference between pre-deployment and post-deployment Offline-RL
>
> The difference is if returns achieved are measured during or after training, i.e. before (as in 2a) or after (as in 2b) deployment.
>
> When tuning hyperparameters, one may only be interested in the performance after tuning. Evaluating policies during tuning is pre-deployment interaction, before the final policy is deployed and evaluated.
>
> When training a recommender system, one deploys after offline training, and starts evaluating. After deployment, one finetunes the system using live user data, requiring post-deployment adaptation.

---

> > ### Comment · Reviewer_7aG4 · 2025-08-05
> >
> > Thanks to the authors for providing detailed explanations about pre-deployment and post-deployment Offline-RL. It would be great to update the clarification in the final version for easier reading!

---

### Official Review · Reviewer_rtSN · 2025-07-02

**Clarity:** 3
**Significance:** 4
**Originality:** 3
**Rating:** 5
**Confidence:** 4

**Summary:**

This work focuses on two primary problems in offline RL: Ambiguous Problem Setting, and Opaque Algorithmic Design. To address these problems, this work introduces a taxonomy of existing offline RL and proposes a unified high-quality implementation of popular offline RL methods. In addition, a combination of algorithmic components is implemented with flexible configuration, called Unifloral, out of which two new methods (TD3-AWR and MoBRAC) are demonstrated to perform well on selected MuJoCo and Adroit tasks from the D4RL suite.

**Questions:**

1. How to understand the “no cost” in “either through pre-deployment interaction, which incurs no cost and occurs before evaluation”? Are the interactions with the environment not counted as a type of cost?
2. How the authors think about the empirical evaluation for zero-shot offline RL (and Offline RL with post-deployment online policy selection)?
3. Will decision transformer and its follow-up methods be scheduled to add to the code base?

**Ethical Concerns:**

["NO or VERY MINOR ethics concerns only"]

**Final Justification:**

I've read the authors' responses and the other reviewers' comments.

After rethinking the contribution of this paper, I think the taxonomy of evaluation protocols and the unified code implementation provided by this work will be useful to the audience in DRL community, especially for those who are interested in offline RL.

I increased my rating accordingly.

**Limitations:**

The limitations are discussed and acknowledged in the checklist.

**Quality:**

3

**Strengths And Weaknesses:**

###Strengthes:

- The revisiting presented in this paper with a new taxonomy and a unified code implementation will be beneficial to RL community.
- The proposed unified code implementation shows an appealing speed-up compared with CORL. The combination implementation Unifloral will be a useful code base for offline RL practitioners to develop upon.
- The simulated pre-deployment policy selection under a given budget offers an effective approach to evaluate offline RL methods in this setting.
- The evaluation results in Figure 3 provide a useful reference. The proposed two new methods are valuable to practitioners.

&nbsp;

###Weaknesses:

- After reading the paper and re-checking the sentences, I am still a bit confused about the difference between pre-deployment interaction and post-deployment adaptation.
    - In Line 105-108, “some methods relax this strict separation by allowing limited interaction with the environment—either through pre-deployment interaction, which incurs no cost and occurs before evaluation, or through post-deployment adaptation, where performance includes the return during these additional interactions.” It seems that the difference is whether the return (i.e., evaluation) is included. But I think it is not the case after reading the following content of the paper. In addition, how to understand the “no cost” in the sentence above?
    - And I found it seems contradictory between “either through pre-deployment interaction, which incurs no cost and occurs before evaluation” (Line 106) and  “Train a set of policies offline, select the best policy based on $N$ online evaluations before deployment” (2a. of the taxonomy). I guess the latter should be the correct one. And I think this confusing expression should be clarified.
- As to the taxonomy,
    - It is also not clear and confusing when reading “(2a.) … select the best policy based on $N$ online evaluations before deployment” and “(2b.) … then deploy one of them. After deployment, select which policy to use based on online performance.” I have no idea about what the exact principle to separate the two categories.
    - Is the “Limited pre-deployment interaction” necessary for Offline-to-Offline RL?
    - Moreover, I suggest the authors make a table to present existing offline RL methods by using this taxonomy.
- I found the “fixed sampling range of each hyperparameter” in “A Definition of Offline RL Methods” somewhat problematic, as the method that uses different ranges can be converted to use a large and fixed range, which seems to fit this definition well.
- For the evaluation and experiments,
    - It seems that only the 2a. in the taxonomy is evaluated for existing methods. I think the pre-deployment selection (with online evaluations) is not necessarily practically feasible in many offline RL problems (This setting suits in-context learning more). Therefore, zero-shot evaluation is still an important category.
    - Line 136, the authors mentioned, “We measure this budget in terms of the number of evaluation episodes”, however, episodes can have different lengths at least in general (i.e., number of timesteps). This does not seem to be a proper measure in this sense. I think this is fine since the considered environments have a pre-defined fixed horizon.

&nbsp;

###Minors

- There are inconsistent formats in reference, e.g., arXiv papers.

---

> ### Author Rebuttal · Authors · 2025-07-30
>
> We thank reviewer rtSN for their questions and criticism of our paper, and we would like to clarify on their remarks.
>
> ## Weaknesses
>
> > After reading the paper and re-checking the sentences, I am still a bit confused about the difference between pre-deployment interaction and post-deployment adaptation.
>
> Recall that we use *deployment* as the point in time at which we stop training and start evaluating our agent. In offline RL, an agent has no access to the environment before deployment. However, some methods relax this strict separation, allowing one (or both) of:
>
> - *pre-deployment interaction*, in which the agent may interact with the environment prior to deployment -- for instance, to select one of several policies to deploy based on online performance. Hyperparameter tuning needs pre-deployment interaction, since policies' returns are used to sample new hyperparameters to test, but the final performance metric is the performance *after* tuning, i.e. after deployment.
>
> - *post-deployment adaptation* allows the agent to continue learning after deployment, and the performance metric includes all returns collected after deployment. For example, when training a recommender system offline and finetuning it with live user data.
>
> We believe that this clarification should address most of your doubts, and have improved the wording in the paper accordingly (including removal of the overloaded word "cost" mentioned in line 106 of the paper).
>
> > As to the taxonomy, \[...\] I have no idea about what the exact principle to separate the two categories \[(2a) and (2b)\].
>
> The answer above should make the difference between categories (2a) and (2b) of the taxonomy clear: (2a) allows for pre-deployment interaction only; (2b) allows for post-deployment adaptation only. The difference is if returns achieved are measured during or after training, i.e. before or after deployment.
>
> > Is the "Limited pre-deployment interaction" necessary for Offline-to-Offline RL?
>
> To answer your question, could you elaborate on what you mean by offline-to-offline RL?
>
> > I found the "fixed sampling range of each hyperparameter" in "A Definition of Offline RL Methods" somewhat problematic, as the method that uses different ranges can be converted to use a large and fixed range, which seems to fit this definition well.
>
> We agree that specifying different hyperparameter ranges for different environments, especially without justification, is problematic. In fact, we explicitly criticise this in line 123 of our paper. Instead, the hyperparameters are part of a method, which should not be environment-specific.
> For fair comparison between methods, we do convert methods that specify multiple ranges into one that specifies one range by taking the ranges' union (line 175).
>
> > For the evaluation and experiments, \[...\] zero-shot evaluation is still an important category.
>
> We note that you can easily compare zero-shot performance by looking at the first datapoint in each of our evaluation plots, as it represents performance after 0 episodes of pre-deployment policy selection.
>
> > episodes can have different lengths at least in general. This does not seem to be a proper measure in this sense.
>
> We argue that both episodes and steps are valid units for budget, both with advantages and drawbacks:
> - Performance in RL is usually measured as episodic return, which makes episodes a natural fit.
> - In practice the cost for evaluating robotics applications is usually dominated by *resetting* the environment (think - placing objects or the robot itself back to their initial positions), which is done once per episode.
> - When setting a budget in steps, one might observe a different number of episodes between policies, and has to compare performances with different confidence levels.
>
> ## Answers to Questions
>
> > How the authors think about the empirical evaluation for zero-shot offline RL (and Offline RL with post-deployment online policy selection)?
>
> As mentioned above, zero-shot evaluation is a special case of our protocol, namely zero pre-deployment interaction episodes.
> We believe that our method can be extended to offline RL with post-deployment by modifying the policy selection via a bandit, but this was beyond the scope of our paper.
>
> > Will decision transformer and its follow-up methods be scheduled to add to the code base?
>
> We have already implemented the decision transformer and present some preliminary results as D4RL normalized scores on three seeds each:
>
> | Dataset | Mean | Stdev |
> | --- | --- | ---  |
> | Hopper-medium-v2  | 5.5 (unstable, max 59.6) | 16.3 |
> | Hopper-medium-expert-v2  | 10.9 (unstable, max 111.05) | 33.1 |
> | Halfcheetah-medium-v2  | 42.4 | 0.5 |
> | Halfcheetah-medium-expert-v2  | 74.6 | 23.1 |
> | Walker2d-medium-expert-v2  | 83.4 | 15.1 |
>
> Thank you for your valuable feedback. Given our response and clarifications to your questions and concerns, as well as the new preliminary results, we kindly ask you to consider increasing your score.

---

> > ### Comment · Reviewer_rtSN · 2025-08-05
> >
> > I appreciate the authors' careful response. The clarifications are helpful to me.
> >
> > > To answer your question, could you elaborate on what you mean by offline-to-offline RL?
> >
> > Sorry for the typo, it should be offline-to-online RL.

---

> > > ### Author Response · Authors · 2025-08-05
> > >
> > > In offline-to-online RL, a policy is first (pre-)trained from offline data before training continues online. Online training assumes (perhaps limited) access to a simulator, while offline data is used as a prior (e.g. to help with complex exploration problems). The measured performance is usually the one after online training, meaning both offline and some online training happen before deployment. Finetuning may continue even after deployment, so offline-to-online RL is the most general case in our taxonomy, including both pre-deployment interaction and post-deployment adaptation.

---

### Official Review · Reviewer_2NhV · 2025-07-03

**Clarity:** 2
**Significance:** 3
**Originality:** 3
**Rating:** 5
**Confidence:** 3

**Summary:**

This paper identifies key issues in offline RL research, including ambiguous problem definitions, inconsistent evaluation practices, and overly complex implementations. To address these, the authors propose a transparent evaluation framework, release minimal single-file implementations of major offline RL methods, and introduce Unifloral, a unified algorithmic framework that spans prior approaches via a shared hyperparameter space. Using this framework, they develop two new algorithms—TD3-AWR and MoBRAC—that outperform standard baselines.

**Questions:**

1. Why is there no discounting in equation 1?
2. The difference between post
4. Why specifically do the authors recommend those particular datasets (line 164)? It seems like the argument is simply "more is better"
5. My interpretation is that "distractor policies" are just policies with higher variance returns (perhaps because they have not yet converged to a nearly deterministic policy)?
4.
> we uniformly sample from the hyperparameter tuning ranges specified in each algorithm’s original paper or the 176 union of ranges when multiple are provided.

Could the authors include these hyperparameter ranges in the appendix? Given that there are many algorithms evaluated, it would be a bit tedious for a reader to track down these values across many papers. This would also help answer one of my questions posed in weaknesses section.
5.
> either through pre-deployment interaction, which incurs no cost and occurs before evaluation, or through post-deployment adaptation, where performance includes the return during 107 these additional interactions.

I'm a bit confused; how does pre-deployment interaction carry no cost? And what does it mean for performance to "include" the return during addition interaction?

**Ethical Concerns:**

["NO or VERY MINOR ethics concerns only"]

**Final Justification:**

As stated in my original review, I think this paper is useful and some of the key results are interesting to me. I believe this paper is a nice contribution to the RL community. Any issues with this paper are minor and easily addressable in the camera-ready.

**Limitations:**

Yes.

**Quality:**

4

**Strengths And Weaknesses:**

# Strengths
1. The paper makes several fair critique of current practice in offline RL research that the community should seriously consider.
1. The framework and empirical evaluation is thorough, capturing a wide range of popular offline RL algorithms.  One result I found particularly interesting that models-based algorithm results reported in prior works are overfitted to the task.
1. The contributed codebase looks promising, and I anticipate it would indeed make it easier to understand what makes or breaks offline RL methods. (Algorithm implementations are indeed opaque. Anecdotally, I distinctly remember having quite a bit of trouble reproducing CQL results on AntMaze tasks years ago. I found that performance hinged on implementation-specific details like reward normalization.)

**I personally feel that the contributions and call to action put forth by this paper outweighs shortcomings and lean accept.**

# Weaknesses

1. **Evaluation procedure.** The proposed evaluation procedure uses random hyperparameter sampling followed by UCB over evaluation rollouts. Why was random sampling chosen over the standard grid search? Additionally, the use of the mean score vs. number-of-evaluations curve may be less relevant in contexts where we care most about the final converged performance. Aren't we ultimately interested in the score these curves converge to? While it’s true that in practice we may be limited in the number of evaluations, doesn't the evaluation curve’s plateau reflects the best estimate of a method's performance?

1. **Is it reasonable to report performance as the aggregate over many hyperparameter settings?** This paper takes the stance that we should factor in hyperparamter tuning as part of the empirical evaluation, which I agree with. But I'm not so sure we should be aggregating performance over hyperparamters during the UCB evaluation. For this to be appropriate, we need to be sure that the range of hyperparameters we use reflects a "reasonable" range [1] -- which is admittedly not well-defined. What happens during evaluation if we set the hyperparameters range to something obviously too wide? This would produce many very suboptimal arms that UCB would quickly eliminate, but we'd also have fewer high-scoring arms which would ultimately lower the score evaluation converges to after collecting a large number of trajectories, right?


# Minor comments

> Thus, the state-of-the-art remains ambiguous, with no method demonstrating uniformly strong performance across all datasets.

This phrasing suggest that if we evaluated algorithms fairly, one SOTA algorithm would rise to the top. But it's common for some methods to perform well on some tasks but not others.

1. Line 186: Missing period/colon after bold header.

---
# References
1. Patterson et al. "Empirical Design in Reinforcement Learning." https://www.jmlr.org/papers/v25/23-0183.html

---

> ### Author Rebuttal · Authors · 2025-07-30
>
> Thank you for appreciating our work's contribution, the thoroughness of our evaluations and the value of our codebase. We address your questions and concerns below.
>
> ## Weaknesses
>
> > The proposed evaluation procedure uses random hyperparameter sampling followed by UCB over evaluation rollouts. Why was random sampling chosen over the standard grid search?
>
> For fair comparison via our bandit protocol, the number of policies trained with each algorithm should be constant. When using grid search however, the number of policies trained would be equal to the number of hyperparameter combinations. Thus, it is necessary to subsample the set of hyperparameter combinations, which we do with random search to remain unbiased.
>
> > Aren't we ultimately interested in the score these curves converge to? While it’s true that in practice we may be limited in the number of evaluations, doesn't the evaluation curve’s plateau reflects the best estimate of a method's performance?
>
> Performance given a limited budget is of very high practical and theoretical significance. For example, zero-shot performance is a practically relevant metric, and siginifcant efforts have been made to maximize zero-shot performance based on offline analysis of trained policies (see e.g. the works referenced in our Related Work section, line 317).
> Our evaluation method gives the flexibility to compare methods at different budgets, **including** zero-shot and final, i.e. plateaued, performance. Thus it is more informative, and measures a method's performance in a more general sense.
>
> > \[...\] we need to be sure that the range of hyperparameters we use reflects a "reasonable" range \[1\] -- which is admittedly not well-defined. What happens during evaluation if we set the hyperparameters range to something obviously too wide? This... would ultimately lower the score evaluation converges to after collecting a large number of trajectories, right?
>
> That is correct - methods with wide hyperparameter ranges, that require a large tuning budget to perform well, will perform poorly under our metric. Our evaluation was designed precisely to demonstrate this point, as prior work considers only the maximum performance within the hyperparameter range, disregarding tuning cost. It is also for this reason that we define offline RL methods (Section 3.1) as including the hyperparameter range, so an improved range over the same hyperparameters is viewed as a superior method.
>
> ## Minor Comments
> > > Thus, the state-of-the-art remains ambiguous, with no method demonstrating uniformly strong performance across all datasets.
> >
> > This phrasing suggest that if we evaluated algorithms fairly, one SOTA algorithm would rise to the top. But it's common for some methods to perform well on some tasks but not others.
>
> We stand by our assessment that SOTA remains ambiguous, since no algorithm has risen to the top. The existance of a single SOTA algorithm is not to be taken for granted, as you have pointed out.
>
> > Line 186: Missing period/colon after bold header
>
> We have fixed that for the camera ready, thank you for pointing it out.
>
> ## Questions
>
> > 1. Why is there no discounting in equation 1?
>
> We consider finite-horizon MDPs, which do not require discounted return in their mathematical formulation as the episodic return is finite. Despite this, many methods still include a discount factor to aid credit assignment, but this is not theoretically necessary.
>
> > 2. The difference between post
>
> We are making an educated guess that this refers to the post and pre-deployment setting; we are happy to clarify further if you were referring to something else.
>
> We use *deployment* as the point in time at which we stop training and start evaluating our agent. In offline RL, an agent has no access to the environment before deployment. However, some methods relax this strict separation, allowing one (or both) of:
>
> - *pre-deployment interaction*, in which the agent may interact with the environment prior to deployment -- for instance, to select one of several policies to deploy based on online performance. Hyperparameter tuning needs pre-deployment interaction, since policies' returns are used to sample new hyperparameters to test, but the final performance metric is the performance *after* tuning, i.e. after deployment.
>
> - *post-deployment adaptation* allows the agent to continue learning after deployment, and the performance metric includes all returns collected after deployment. For example, when training a recommender system offline and finetuning it with live user data.
>
> We believe that this clarification addresses most of the reviewers' doubts, and have improved the wording in the paper accordingly (including the usage of the overloaded word "cost" mentioned in line 106 of the paper).
>
>
> > 3. Why specifically do the authors recommend those particular datasets (line 164)? It seems like the argument is simply "more is better"
>
> We select a fixed set of datasets to make future evaluations consistent and immune to cherry-picking. Of these, the individual datasets were selected to provide coverage over the maximum number of environments and behavior policies, as well as focusing *only* on datasets where existing methods do not achieve trivial performance (contantly nil or perfect), providing meaningful signal for future algorithm comparison. We note that for halfcheetah and walker datasets, the bandit evaluations converged to a clear final ranking that is different from other intermediate points throughout the bandit's score trajectory. This further demonstrates the additional insight our proposed evaluation procedure provides.
>
> > 4. My interpretation is that "distractor policies" are just policies with higher variance returns (perhaps because they have not yet converged to a nearly deterministic policy)?
>
> You're correct that distractor policies have higher variance, however, it is typically assumed that this is due to an increased number of low-performance episodes. Instead, we show that this is also due to a *higher maximum performance*, which consistently high-performing policies fail to achieve. This is a presents challenges for bandit sampling, causing bandits to initially *prefer* distractor policies, as shown in Appendix D.
>
> > 5. Could the authors include these hyperparameter ranges in the appendix? Given that there are many algorithms evaluated, it would be a bit tedious for a reader to track down these values across many papers. This would also help answer one of my questions posed in weaknesses section.
>
> We use ranges from each algorithms's original paper, as described in line 175.
> Our algorithm's hyperparameter ranges can be found in appendix H. This presents in the unified space of hyperparameters, allowing for direct comparison of hyperparameters across methods, a feature missing from previous work.
>
> > 6. I'm a bit confused; how does pre-deployment interaction carry no cost? And what does it mean for performance to "include" the return during addition interaction?
>
> The performance of an agent before deployment does not affect the final evaluation metric. For example, you might test a few policies on a robot, while only measuring the performance of the best one (which you deploy).
> We have changed wording in the camera ready version to correctly use the word "cost" in the context of rolling out episodes, rephrasing the relevant sentence.

---

> > ### Comment · Reviewer_2NhV · 2025-08-05
> >
> > Thank you for the detailed response! I still think this paper would be a useful contribution to the community and will maintain my score.
> >
> > A few followup comments:
> > 1. My incomplete "difference between post" question was referring to post- and pre-deployment setting (sorry about that). The phrasing in your response clarifies this distinction; I think the use of the term "cost" was throwing me off a bit.
> > 2. Your clarification regarding the distractor policies helps a lot. It's an interesting observation and provides further support for the evaluation procedure proposed here.

---

### Note · Authors · 2025-08-13

Dear Reviewers, AC, and SAC,

We are grateful for the productive discussion period and thank the reviewers for their encouraging feedback on our work's contribution and potential impact.

To summarise the discussion:

* Reviewers were unanimous about our work’s quality (scores: 4,3,4,3) and significance (scores: 3,4,4,3).
* The discussion successfully resolved the primary question raised regarding our taxonomy. In response to feedback, we refined the definitions for variants 2a and 2b, and the reviewers (rtSN, eMD6) confirmed that this revision addresses their concerns.
* Finally, we added new results for an additional algorithm (Decision Transformer), showcasing our framework’s capacity for rapid evaluation.

Thank you again for your time and valuable feedback.

---

### Decision · Program_Chairs · 2025-09-17

**Decision:**

Accept (oral)

**Comment:**

This work makes a contribution to the research practice in offline RL by identifying several problematic issues across empirical offline RL papers: ambiguous problem definitions, inconsistent algorithm designs and implementations, and weak, biased evaluations. The paper addresses these by introducing a clean taxonomy, evaluation protocol, and, crucially, implementations of a variety of RL algorithms, as well as a unifying algorithmic framework, Unifloral. The paper also presents two algorithms within this framework that outperform the alternatives.

The metareviewer and the reviewers believe that this is an elegant work that does a lot for making offline RL research results more reliable and comparable to each other, which promises to have a major positive impact on the field.